# Un-Mixing Test-Time Normalization Statistics: Combatting Label Temporal Correlation

**Devavrat Tomar**[1*]    **Guillaume Vray**[2*]    **Jean-Philippe Thiran**[1,2]    **Behzad Bozorgtabar**[1,2]

[1]EPFL    [2]CHUV

{firstname}.{lastname}@epfl.ch

## Abstract

Recent test-time adaptation methods heavily rely on nuanced adjustments of batch normalization (BN) parameters. However, one critical assumption often goes overlooked: that of independently and identically distributed (i.i.d.) test batches with respect to unknown labels. This oversight leads to skewed BN statistics and undermines the reliability of the model under non-i.i.d. scenarios. To tackle this challenge, this paper presents a novel method termed 'Un-Mixing Test-Time Normalization Statistics' (UnMix-TNS). Our method re-calibrates the statistics for each instance within a test batch by *mixing* it with multiple distinct statistics components, thus inherently simulating the i.i.d. scenario. The core of this method hinges on a distinctive online *unmixing* procedure that continuously updates these statistics components by incorporating the most similar instances from new test batches. Remarkably generic in its design, UnMix-TNS seamlessly integrates with a wide range of leading test-time adaptation methods and pre-trained architectures equipped with BN layers. Empirical evaluations corroborate the robustness of UnMix-TNS under varied scenarios—ranging from single to continual and mixed domain shifts, particularly excelling with temporally correlated test data and corrupted non-i.i.d. real-world streams. This adaptability is maintained even with very small batch sizes or single instances. Our results highlight UnMix-TNS's capacity to markedly enhance stability and performance across various benchmarks. Our code is publicly available at https://github.com/devavratTomar/unmixtns.

## 1    Introduction

Deep neural networks (DNNs), when deployed in real-world test environments, often face the pervasive challenge of domain shift, potentially undermining their performance. Addressing this, the research community has advanced towards the forefront of online test-time adaptation (TTA). This involves a myriad of methodologies aimed at recalibrating the batch normalization (BN) layers (Ioffe & Szegedy, 2015), a cornerstone of deep learning architectures informed by real-time test data. BN layers play a critical role in stabilizing the training process and enhancing model generalization. These TTA approaches function by either re-estimating normalization statistics based on the present test batch (Nado et al., 2020; Schneider et al., 2020; Benz et al., 2021) or additionally fine-tuning the BN parameters to minimize test-time losses, such as those resulting from entropy minimization (Wang et al., 2020). Specifically, the former has concentrated on addressing the performance decline observed in conventional BN when subjected to domain shifts during testing. This diminution in model efficacy on previously unseen test data is primarily ascribed to shifts in the statistical properties of intermediate layers relative to those conserved from the source training dataset. Intrinsically, BN marginalizes inconsequential instance-wise variations by decorrelating feature sets across batches, assuming that these batches are uniformly populated with samples from diverse categories. If test batches are also uniformly sampled from different categories, employing TTA methods that renormalize features based on immediate statistics from the current test batch can counteract the domain-induced distribution shifts.

Nevertheless, these methods come with their own set of challenges and assumptions. Typically, they operate under the assumption of large test batch sizes and a singular, static distribution shift.

---

*denotes equal contribution.

Moreover, these methods generally presume that the test batches are independently and identically distributed (i.i.d.) concerning their true labels. This i.i.d. assumption, though useful for simplification, frequently does not hold true under real-world scenarios. Take the case of autonomous driving, where a variety of diverse and unpredictable factors makes it improbable for incoming test batches to conform to an i.i.d. distribution. In such contexts, conventional BN-based TTA methods fall short, producing unstable and unreliable adaptations. To overcome this issue, recent methods (Gong et al., 2022; Yuan et al., 2023) have introduced the concept of a balanced, pseudo-label-based memory bank. This memory bank serves as a repository for test images, facilitating the online estimation of unbiased BN statistics and the integration of instance-level feature statistics with those derived from source data. While promising, their utility is often limited to particular situations. Notably, they falter in scenarios where privacy concerns curtail data retention. Furthermore, choosing the optimal weight hyperparameter for merging instance-level statistics with pre-existing batch statistics can introduce a layer of computational overhead post-deployment, complicating the model's adaptability. In contrast, methodologies such as (Niu et al., 2023) and (Marsden et al., 2024) have advocated for instance-level normalization techniques, such as (Ba et al., 2016) as viable alternatives to BN for training on source data. Other alternatives exhibit increased robustness to varying batch size and long-tailed distribution via group normalization (Wu & He, 2018), Compound Batch Normalization (Cheng et al., 2022), and Mixture Normalization (Kalayeh & Shah, 2019).

In this paper, we thoroughly revisit BN for test-time adaptation, targeting temporally correlated (non-i.i.d.), distributionally shifted test batches. In our approach, we interpret the instance-wise input features of BN layers pertaining to the present test batch as samples drawn in non-i.i.d. manner from $K$ distinct distributions over time, reflecting temporal correlation. Consequently, we *unmix* the initially stored batch normalization statistics into $K$ distinctive components, each reflecting statistics from similar test inputs. This unveils a strategy for real-time adaptation of these statistics to the streaming test batches. Drawing inspiration from sequential clustering paradigms, our method aims to update the $K$ statistics components using the closest instances from the streaming test data in a dynamic online setting. In summary, our contributions are as follows:

- We introduce a novel test-time normalization scheme (UnMix-TNS) designed to withstand the challenges posed by label temporal correlation of test data streams.

- UnMix-TNS demonstrates robustness across various test-time distribution shifts such as single domain, and continual domain. While not primarily designed for mixed domain settings, it offers a level of adaptability in these scenarios. Additionally, the method excels with small batch sizes, even down to individual samples.

- UnMix-TNS layers, with negligible inference latency, seamlessly integrate into pre-trained neural network architectures, fortified with BN layers, boosting test-time adaptation capabilities while incurring minimal overhead.

- Our results demonstrate notable enhancements in TTA methods under non-i.i.d. conditions when augmented with UnMix-TNS, as evidenced on datasets involving corruptions and natural shifts. We also unveil ImageNet-VID-C and LaSOT-C video datasets, corrupted versions of ImageNet-VID and LaSOT, for realistic domain shift analysis in video classification.

## 2 METHODOLOGY

### 2.1 PRELIMINARIES

**Batch normalization in TTA.**    In addressing the challenges of covariate shifts at test time, test-time BN (TBN) employs the current batch's statistics rather than relying on stored source statistics. Consider a batch of feature maps being input into the BN layer. We can represent this batch as $\mathbf{z} \in \mathbb{R}^{B \times C \times L}$, where $B$ stands for the batch size, $C$ signifies the number of channels, and $L$ denotes the dimensions of each feature map. The BN layer normalizes the feature maps using the current batch statistics, denoted as $(\mu^t, \sigma^t) \in \mathbb{R}^C$. Following this, it employs the affine parameters $(\gamma, \beta) \in \mathbb{R}^C$ to produce normalized feature maps $\hat{\mathbf{z}}$ that are both scaled and shifted, as follows:

$$\hat{\mathbf{z}}_{:,c,:} = \gamma_c \cdot \frac{\mathbf{z}_{:,c,:} - \mu_c^t}{\sigma_c^t} + \beta_c, \quad \mu_c^t = \frac{1}{BL} \sum_{b,l} \mathbf{z}_{b,c,l}, \quad \sigma_c^t = \sqrt{\frac{1}{BL} \sum_{b,l} (\mathbf{z}_{b,c,l} - \mu_c^t)^2}, \quad (1)$$

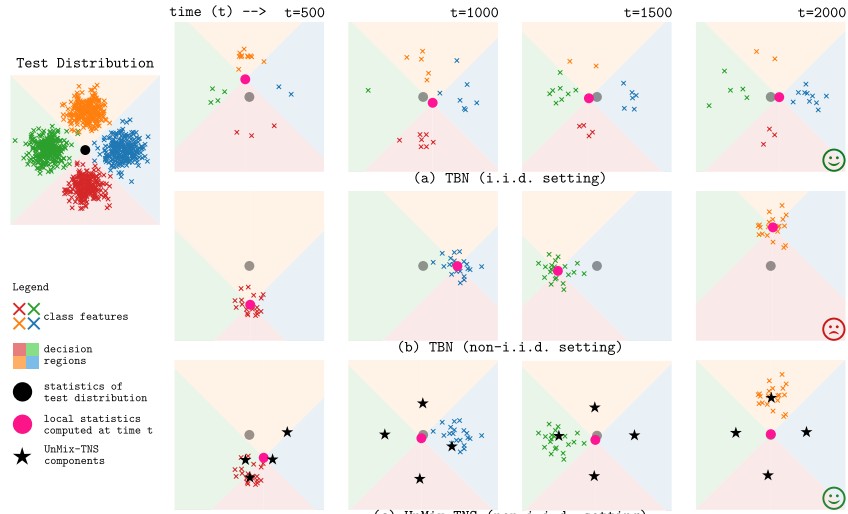

Figure 1: **Test-Time BN (TBN) vs. UnMix-TNS**. (a) TBN recalibrates its intermediate features when test batches are i.i.d. sampled over time $t$, accommodating distribution shifts. (b) However, TBN fails for non-i.i.d. label-based test batch sampling, leading to skewed batch statistics. (c) UnMix-TNS overcomes this failure by estimating unbiased batch statistics through its $K$ statistics components.

**Test-time adaptation under label temporal correlation.** Let $f_\theta : \mathcal{X} \to \mathcal{Y}$ denote a neural network parameterized by $\theta$, mapping image space $\mathcal{X}$ to label space, $\mathcal{Y}$. This network, featuring BN layers, has been trained on the source data distribution $P_\mathcal{S}(\mathbf{x}, y)$. Given a stream of covariate shifted test images $\mathbf{x_t}$ sampled at time $t$ from an arbitrary test distribution $P_\mathcal{T}(\mathbf{x}, y|t)$ with temporally correlated labels, the goal is to continuously adapt $f_\theta$ to new data $\mathbf{x}_t$, as it arrives, even without access to the corresponding true label $y_t$.

## 2.2 UNMIXING TEST-TIME NORMALIZATION STATISTICS

In Figure 1, a toy example clearly elucidates a crucial point: when dealing with covariate-shifted target test images that bear temporal correlations, there's an intrinsic bias in estimating online batch normalization statistics, as depicted in Figure 1(b). This bias is in sharp contrast to the more stable dynamics of batch normalization statistics sourced from i.i.d. batches of well-shuffled data, as visualized in Figure 1(a). This variance can lead to substantial failures in many test-time adaptation methods, particularly when test images from the target domain exhibit temporal correlations tied to their true labels.

This section introduces UnMix-TNS, our proposed normalization paradigm tailored for the non-i.i.d. streams of test images. As illustrated in Figure 1(c), at the heart of the UnMix-TNS layer is a process that deftly decomposes the stored BN statistics into $K$ components. Then, all $K$ statistics components are updated based on their alignment with the test batch's instance-wise statistics. Components that align more closely undergo more substantial updates, while others reflect statistics from previously encountered, less similar features, simulating an ideal i.i.d. environment. The genius lies in the end result: the statistics computed by UnMix-TNS with its $K$ components in non-i.i.d. conditions, align seamlessly with those generated by TBN in i.i.d. conditions (see Appendix A.3 for theoretical analysis).

In the subsequent Section 2.2.1, we formulate the distribution of $K$ UnMix-TNS components and describe how to compute the label unbiased normalization statistics at a temporal instance $t$, utilizing only the statistics derived from the $K$ components in conjunction with the current statistics of the non-i.i.d. test batch. In Section 2.2.2, we explain the process of initializing the $K$ statistics components of the UnMix-TNS layer, leveraging the batch normalization statistics stored within the provided pre-trained model. In Section 2.2.3, we outline the methodology for deriving current statistics from these $K$ statistics components. This process is crucial for both normalizing the input test features and their dynamic online adaptation.

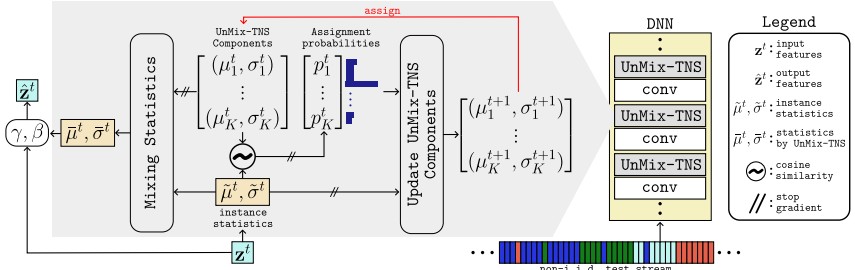

Figure 2: **An Overview of UnMix-TNS.** Given a batch of non-i.i.d test features $\mathbf{z^t} \in \mathbb{R}^{B \times C \times L}$ at a temporal instance $t$, we mix the instance-wise statistics $(\tilde{\mu}^t, \tilde{\sigma}^t) \in \mathbb{R}^{B \times C}$ with $K$ UnMix-TNS components. The alignment of each sample in the batch with the UnMix-TNS components is quantified through similarity-derived assignment probabilities $p_k^t$. This aids both the *mixing* process and subsequent component updates for time $t + 1$.

### 2.2.1 Distribution of $K$ UnMix-TNS Components Mixture

Let $[\mu_1^t, \ldots, \mu_K^t]$ and $[\sigma_1^t, \ldots, \sigma_K^t]$ denote the statistics of the $K$ components within the UnMix-TNS layer at a given temporal instance $t$, where each $\mu_k^t, \sigma_k^t \in \mathbb{R}^C$. We articulate the distribution $h_Z^t(z)$ of instance-wise test features $z \in \mathbb{R}^C$ marginalized over all labels at a temporal instance $t$ using the $K$ components:

$$h_{Z_c}^t(z_c) = \frac{1}{K} \sum_k \mathcal{N}(\mu_{k,c}^t, \sigma_{k,c}^t), \tag{2}$$

where $\mathcal{N}$ represents the Normal distribution function. Given only $(\mu_k^t, \sigma_k^t)_{k=1}^K$, one can derive the label unbiased normalization test statistics $(\bar{\mu}^t, \bar{\sigma}^t)$ at time $t$ (see Appendix A.1) as follows:

$$\bar{\mu}^t = \mathbb{E}_{h_{Z_c}^t}[z_c] = \frac{1}{K} \sum_k \mu_{k,c}^t, \tag{3}$$

$$(\bar{\sigma}^t)^2 = \mathbb{E}_{h_{Z_c}^t}[(z_c - \bar{\mu}^t)^2] = \frac{1}{K} \sum_k (\sigma_{k,c}^t)^2 + \frac{1}{K} \sum_k (\mu_{k,c}^t)^2 - \left(\frac{1}{K} \sum_k \mu_{k,c}^t\right)^2, \tag{4}$$

Subsequent sections will delve into the initialization scheme for the $K$ UnMix-TNS components and elucidate the process of updating their individual statistics $(\mu_k^t, \sigma_k^t)$ at temporal instance $t$.

### 2.2.2 Initializing UnMix-TNS Components

Consider the pair $(\mu, \sigma) \in \mathbb{R}^C$, which stands as the stored means and standard deviation of the features in a given BN layer of $f_\theta$, derived from the source training dataset before adaptation. We initialize statistics $(\mu_k^t, \sigma_k^t)$ of $K$ components of UnMix-TNS at $t = 0$, as delineated in Equation (5).

$$\mu_{k,c}^0 = \mu_c + \sigma_c \sqrt{\frac{\alpha \cdot K}{K - 1}} \cdot \zeta_{k,c}, \quad \sigma_{k,c}^0 = \sqrt{1 - \alpha} \cdot \sigma_c, \quad \zeta_{k,c} \sim \mathcal{N}(0, 1) \tag{5}$$

where $\zeta_{k,c}$ is sampled from the standard normal distribution $\mathcal{N}(0, 1)$, and $\alpha \in (0, 1)$ is a hyperparameter that controls the extent to which the means of the UnMix-TNS components deviate from the stored means of BN layer during initialization. Note that the initial normalization statistics $(\bar{\mu}^0, \bar{\sigma}^0)$ estimated from $(\mu_k^0, \sigma_k^0)$ have an expected value equal to the stored statistics of the BN layer, i.e., $\mathbb{E}[\bar{\mu}_k^0] = \mu$ and $\mathbb{E}[(\bar{\sigma}_k^0)^2] = \sigma^2$ for all values of $\alpha$. Further insights into this relationship are detailed in Appendix A.2.

### 2.2.3 Redefining Feature Normalization through UnMix-TNS

Considering the current batch of temporally correlated features $\mathbf{z}^t \in \mathbb{R}^{B \times C \times L}$, we commence by calculating the instance-wise means and standard deviation $(\tilde{\mu}^t, \tilde{\sigma}^t) \in \mathbb{R}^{B \times C}$, mirroring the Instance Normalization Ulyanov et al. (2016), i.e.:

$$\tilde{\mu}_{b,c}^t = \frac{1}{L} \sum_l \mathbf{z}_{b,c,l}^t, \quad \tilde{\sigma}_{b,c}^t = \sqrt{\frac{1}{L} \sum_l (\mathbf{z}_{b,c,l}^t - \tilde{\mu}_{b,c}^t)^2}, \tag{6}$$

To gauge the likeness between UnMix-TNS components and current test features, we compute the cosine similarity $s_{b,k}^t$ of the current instance-wise means $\tilde{\mu}_{b,:}^t$ with that of $K$ BN components $\{\mu_k^t\}_{k=1}^K$ as follows:

$$s_{b,k}^t = \texttt{sim}\left(\tilde{\mu}_{b,:}^t, \mu_k^t\right), \tag{7}$$

where $\texttt{sim}(\mathbf{u}, \mathbf{v}) = \frac{\mathbf{u}^T \mathbf{v}}{\|\mathbf{u}\|\|\mathbf{v}\|}$ denote the dot product between $l_2$ normalized $\mathbf{u}$ and $\mathbf{v} \in \mathbb{R}^C$. We proceed to derive the refined feature statistics, denoted as $(\bar{\mu}^t, \bar{\sigma}^t) \in \mathbb{R}^{B \times C}$ for each instance. This is accomplished by a weighted mixing of the current instance statistics, $(\tilde{\mu}^t, \tilde{\sigma}^t)$, harmoniously blended with the $K$ BN statistics components, $(\mu_k^t, \sigma_k^t)_{k=1}^K$. This mixing strategy unfolds similar to Equation (3) as:

$$\bar{\mu}_{b,c}^t = \frac{1}{K} \sum_k \hat{\mu}_{b,k,c}^t, \tag{8}$$

$$(\bar{\sigma}_{b,c}^t)^2 = \frac{1}{K} \sum_k (\hat{\sigma}_{b,k,c}^t)^2 + \frac{1}{K} \sum_k (\hat{\mu}_{b,k,c}^t)^2 - (\frac{1}{K} \sum_k \hat{\mu}_{b,k,c}^t)^2, \tag{9}$$

$$\hat{\mu}_{b,k,c}^t = (1 - p_{b,k}^t) \cdot \mu_{k,c}^t + p_{b,k}^t \cdot \tilde{\mu}_{b,c}^t, \quad (\hat{\sigma}_{b,k,c}^t)^2 = (1 - p_{b,k}^t) \cdot (\sigma_{k,c}^t)^2 + p_{b,k}^t \cdot (\tilde{\sigma}_{b,c}^t)^2, \tag{10}$$

In this formulation, $p_{b,k}^t = \frac{\exp(s_{b,k}^t/\tau)}{\sum_\kappa \exp(s_{b,k}^t/\tau)}$ represents the assignment probability of the $b^{th}$ instance's statistics in the batch belonging to the $k^{th}$ statistics component. Note that, $p_{b,k}^t \approx 1$ if $b^{th}$ instance exhibits a greater affinity to the $k^{th}$ component, and vice-versa. We employ the refined feature statistics $(\bar{\mu}^t, \bar{\sigma}^t)$ to normalize $\mathbf{z}^t$, yielding normalized features $\hat{\mathbf{z}}^t$ elaborated upon below:

$$\hat{\mathbf{z}}_{b,c,:}^t = \gamma_c \cdot \frac{\mathbf{z}_{b,c,:}^t - \bar{\mu}_{b,c}^t}{\bar{\sigma}_{b,c}^t} + \beta_c, \tag{11}$$

In the concluding steps, all $K$ BN statistics components undergo refinement. This is achieved by updating them based on the weighted average difference between the current instance statistics, with weights drawn from the corresponding assignment probabilities across the batch:

$$\mu_{k,c}^{t+1} \leftarrow \mu_{k,c}^t + \frac{\lambda}{B} \sum_b p_{b,k}^t \cdot (\tilde{\mu}_{b,c}^t - \mu_{k,c}^t), \tag{12}$$

$$(\sigma_{k,c}^{t+1})^2 \leftarrow (\sigma_{k,c}^t)^2 + \frac{\lambda}{B} \sum_b p_{b,k}^t \cdot ((\tilde{\sigma}_{b,c}^t)^2 - (\sigma_{k,c}^t)^2), \tag{13}$$

where $\lambda$ is the momentum hyperparameter and is set based on the principles of momentum batch normalization as proposed by (Yong et al., 2020) (see Appendix A.4). The more a statistic component is closely aligned with the instance-wise statistics in the test batch (precisely when $p_{b,k}^t \approx 1$), the more it undergoes a substantial update.

## 3 EXPERIMENTS

### 3.1 EXPERIMENTAL SETUP

To maintain fairness in comparison with baselines, our experiments are carried out using the open-source online TTA repository (Marsden et al., 2024), which amasses source codes and configurations of state-of-the-art TTA methods. **Implementation details are elaborated in Appendix B.1.**

**Datasets and models.** To examine the repercussions of common corruption, we use the Robust-Bench benchmark (Hendrycks & Dietterich, 2019). This benchmark encapsulates the CIFAR10-C, CIFAR100-C, and large-scale ImageNet-C datasets, offering a comprehensive view of 15 diverse corruption types, each implemented at five distinct severity levels—applied to the corresponding clean test datasets. Our chosen backbone models consist of WideResNet-28 (Zagoruyko & Komodakis, 2016), ResNeXt-29 (Xie et al., 2017), and ResNet-50 (He et al., 2016), each trained on CIFAR10, CIFAR100, and ImageNet, respectively. To assess the robustness of our method to natural domain shifts, we utilize the subset of the DomainNet dataset (Peng et al., 2019), renowned for its pronounced domain shifts, focusing on classification tasks. Given the presence of noisy labels in the original

DomainNet dataset, we adopt the approach from (Chen et al., 2022) and utilize a refined subset, DomainNet-126 (Peng et al., 2019), which features 126 classes across four distinct domains: Real, Sketch, Clipart, and Painting. For every domain, a single ResNet-101 model is trained following the (Chen et al., 2022) and, subsequently, evaluated against the remaining three domains. Furthermore, to evaluate our model's resilience in non-i.i.d. video frame-wise classification contexts, we introduce corrupted versions of ImageNet-VID (Russakovsky et al., 2015) and LaSOT (Dave et al., 2020), named ImageNet-VID-C and LaSOT-C, respectively. Each dataset is modified with three types of corruptions: Gaussian noise, artificial snow, and rain, to challenge the models further. As a backbone, we use ResNet-50 pre-trained on an original, uncorrupted training set of ImageNet.

**Baselines.** In our evaluation, we have benchmarked UnMix-TNS against a diverse range of other test-time normalization methods. Our comparative analysis includes test-time BN recalibration approach (TBN) (Nado et al., 2020), $\alpha$-BN (You et al., 2021), the Instance Aware BN (IABN) layer introduced in NOTE (Gong et al., 2022), and the Robust BN (RBN) layer proposed in RoTTA (Yuan et al., 2023). Beyond this, we investigate the advantages of pairing UnMix-TNS with different test-time optimization methods, comparing it to standard BN layer usage. In our assessment, we explore several TTA methods, including TENT (Wang et al., 2020), which leverages entropy minimization to fine-tune BN affine parameters, and CoTTA (Wang et al., 2022), which optimally employs the Mean teacher method. Additionally, we examine the parameter-free strategy, LAME (Boudiaf et al., 2022), which adjusts model outputs based on batch predictions. As for TTA methods utilizing memory banks to simulate i.i.d. samples, we assess NOTE (Gong et al., 2022) and RoTTA (Yuan et al., 2023). We also explore the more recent universal TTA method, ROID (Marsden et al., 2024), incorporating various regularization techniques, including diversity weighting.

**Evaluation protocols.** All experiments are conducted in an online non-i.i.d TTA setting, with immediate evaluations of predictions. Following the non-i.i.d protocols outlined in (Marsden et al., 2024), we first explore the *single* domain adaptation scenario, wherein the model sequentially adapts to each domain, with a reset in weights upon switching domains. Next, we examine the *continual* domain adaptation scenario, allowing the model to adapt sequentially across all domains without weight reset. Lastly, we assess the *mixed* domain adaptation scenario, evaluating performance with test batches composed of examples from multiple domains. This approach enables a concise yet comprehensive analysis of model adaptability in diverse domains.

## 3.2 RESULTS

Tables 1 and 2 present the average online classification error rates for our method and baselines on corruption and natural shift benchmarks, following three outlined evaluation protocols. Key observations include:

Table 1: **Non-i.i.d. test-time adaptation on corruption benchmarks.** Averaged online classification error rate (in %) across 15 corruptions at severity level 5 on CIFAR10-C, CIFAR100-C, and ImageNet-C, comparing *single*, *continual*, and *mixed* domain adaptation. Averaged over three runs.

| Dataset | SINGLE DOMAIN | | | | CONTINUAL DOMAIN | | | | MIXED DOMAIN | | | |
|---|---|---|---|---|---|---|---|---|---|---|---|---|
| | CIFAR10-C | CIFAR100-C | ImageNet-C | Mean | CIFAR10-C | CIFAR100-C | ImageNet-C | Mean | CIFAR10-C | CIFAR100-C | ImageNet-C | Mean |
| Source | 43.5 | 46.5 | 82.0 | 57.3 | 43.5 | 46.5 | 82.0 | 57.3 | 43.5 | 46.5 | 82.0 | 57.3 |
| **TEST-TIME NORMALIZATION** | | | | | | | | | | | | |
| TBN | 76.0 | 81.6 | 83.2 | 80.3 | 76.0 | 81.6 | 83.2 | 80.3 | 79.2 | 94.3 | 96.6 | 90.0 |
| $\alpha$-BN | 44.4 | 50.7 | 75.7 | 56.9 | 44.4 | 50.7 | 75.7 | 56.9 | 53.0 | 64.4 | 86.7 | 68.0 |
| IABN | 29.1 | 55.7 | 85.0 | 56.6 | 29.1 | 55.7 | 85.0 | 56.6 | **29.1** | 55.7 | 85.0 | **56.6** |
| RBN | 54.3 | 44.6 | 71.3 | 56.7 | 54.7 | 44.9 | 71.3 | 57.0 | 77.3 | 82.4 | 91.1 | 83.6 |
| **UnMix-TNS** | **27.0** | **39.2** | **70.6** | **45.6** | **26.8** | **39.2** | **70.4** | **45.5** | 41.9 | **50.1** | **84.3** | 58.8 |
| **TEST-TIME ADAPTATION** | | | | | | | | | | | | |
| TENT | 76.0 | 81.6 | 82.6 | 80.1 | 75.9 | 81.9 | 82.0 | 79.9 | 79.1 | 94.7 | 97.6 | 90.5 |
| **(+UnMix-TNS)** | 27.0 (-49.0) | 38.7 (-42.9) | 69.5 (-13.1) | 45.1 (-35.0) | 26.6 (-49.3) | 37.7 (-44.2) | 88.2 (+6.2) | 50.8 (-29.1) | 38.4 (-40.7) | 51.2 (-43.5) | 95.0 (-2.6) | 61.5 (-29.0) |
| CoTTA | 77.2 | 80.9 | 82.7 | 80.3 | 77.8 | 81.2 | 82.6 | 80.5 | 81.4 | 94.3 | 96.7 | 90.8 |
| **(+UnMix-TNS)** | 49.1 (-28.1) | 50.1 (-30.8) | 70.7 (-12.0) | 56.6 (-23.7) | 44.6 (-33.2) | 50.4 (-30.8) | 71.4 (-11.2) | 55.5 (-25.0) | 72.1 (-9.3) | 66.6 (-27.7) | 85.6 (-11.1) | 74.8 (-16.0) |
| LAME | 30.6 | 35.2 | 79.3 | 48.4 | 30.6 | 35.2 | 79.3 | 48.4 | 16.1 | **3.8** | **65.0** | 28.3 |
| **(+UnMix-TNS)** | **5.4** (-25.2) | 31.7 (-3.5) | **64.1** (-15.2) | 33.7 (-14.7) | **8.0** (-22.6) | 30.6 (-4.6) | **63.8** (-15.5) | 34.1 (-14.3) | **8.0** (-8.1) | 4.2 (+0.4) | 68.5 (+3.5) | **26.9** (-1.4) |
| NOTE | 26.0 | 53.4 | 81.7 | 53.7 | 26.7 | 53.8 | 81.8 | 54.1 | 36.1 | 57.0 | 85.6 | 59.6 |
| **(+UnMix-TNS)** | 26.7 (+0.7) | 38.3 (-15.1) | 70.9 (-10.8) | 45.3 (-8.4) | 26.7 (+0.0) | 38.5 (-15.3) | 70.8 (-11.0) | 45.3 (-8.8) | 52.6 (+16.5) | 53.8 (-3.2) | 84.6 (-1.0) | 63.7 (+4.1) |
| RoTTA | 27.7 | 43.5 | 70.2 | 47.1 | 27.9 | 44.3 | 68.9 | 47.0 | 64.0 | 65.0 | 86.7 | 71.9 |
| **(+UnMix-TNS)** | 26.8 (-0.9) | 39.4 (-4.1) | 70.8 (+0.6) | 45.7 (-1.4) | 26.8 (-1.1) | 38.9 (-5.4) | 69.6 (+0.7) | 45.1 (-1.9) | 45.4 (-18.6) | 53.2 (-11.8) | 83.9 (-2.8) | 60.8 (-11.1) |
| ROID | 73.4 | 77.7 | 82.4 | 77.8 | 73.4 | 77.7 | 82.4 | 77.8 | 77.3 | 93.5 | 96.5 | 89.1 |
| **(+UnMix-TNS)** | 15.3 (-58.1) | **14.0** (-63.7) | 66.6 (-15.8) | **32.0** (-45.8) | 15.5 (-57.9) | **13.4** (-64.3) | 66.4 (-16.0) | **31.8** (-46.0) | 32.6 (-44.7) | 12.3 (-81.2) | 77.5 (-19.0) | 40.8 (-48.3) |

**UnMix-TNS exemplifies resilience under non-i.i.d. test-time adaptation.** Our observation reveals that, in comparison to other test-time normalization-based methods, UnMix-TNS stands superior, delivering exceptional average classification performance in both *single* and *continual* domain

Table 2: **Non-i.i.d. test-time adaptation on natural shift benchmark (DomainNet-126).** Online classification error rate (in %) depicted for each source domain, averaged across the other target domains. A comparative analysis between *single*, *continual*, and *mixed* domain adaptation is presented. Averaged over three runs.

| | SINGLE DOMAIN | | | | | CONTINUAL DOMAIN | | | | | MIXED DOMAIN | | | | |
|---|---|---|---|---|---|---|---|---|---|---|---|---|---|---|---|
| Source domain | clipart | painting | real | sketch | Mean | clipart | painting | real | sketch | Mean | clipart | painting | real | sketch | Mean |
| Source | 49.5 | 41.6 | 45.2 | 45.3 | 45.4 | 49.5 | 41.6 | 45.2 | 45.3 | 45.4 | 49.5 | 41.6 | 45.2 | 45.3 | 45.4 |
| *TEST-TIME NORMALIZATION* | | | | | | | | | | | | | | | |
| TBN | 89.2 | 87.7 | 85.8 | 87.4 | 87.5 | 89.2 | 87.7 | 85.8 | 87.4 | 87.5 | 93.7 | 93.0 | 89.9 | 92.9 | 92.4 |
| α-BN | 60.7 | 53.9 | 58.1 | 55.4 | 57.0 | 60.7 | 53.9 | 58.1 | 55.4 | 57.0 | 64.7 | 60.5 | 61.7 | 59.5 | 61.6 |
| IABN | 69.2 | 62.9 | 67.2 | 65.3 | 66.2 | 69.2 | 62.9 | 67.2 | 65.3 | 66.2 | 69.2 | 62.9 | 67.2 | 65.3 | 66.2 |
| RBN | 66.6 | 61.1 | 61.2 | 59.9 | 62.2 | 66.6 | 61.2 | 61.2 | 59.9 | 62.2 | 79.9 | 76.6 | 71.3 | 76.3 | 76.0 |
| **UnMix-TNS** | 48.9 | 42.9 | 48.8 | 41.5 | 45.5 | 48.0 | 41.6 | 47.7 | 40.6 | 44.5 | 48.7 | 41.6 | 47.6 | 39.5 | 44.4 |
| *TEST-TIME ADAPTATION* | | | | | | | | | | | | | | | |
| TENT | 89.2 | 87.7 | 85.8 | 87.4 | 87.5 | 89.2 | 87.7 | 85.9 | 87.4 | 87.5 | 93.7 | 93.1 | 89.9 | 92.9 | 92.4 |
| (+UnMix-TNS) | 48.8 (-40.4) | 42.7 (-45.0) | 48.7 (-37.1) | 41.5 (-45.9) | 45.4 (-42.1) | 47.9 (-41.3) | 41.5 (-46.2) | 47.5 (-38.4) | 40.6 (-46.8) | 44.4 (-43.1) | 48.1 (-45.6) | 41.5 (-51.6) | 47.0 (-42.9) | 39.2 (-53.7) | 43.9 (-48.5) |
| CoTTA | 89.2 | 87.7 | 85.8 | 87.4 | 87.5 | 89.2 | 87.7 | 85.8 | 87.4 | 87.5 | 93.7 | 93.1 | 89.9 | 92.9 | 92.4 |
| (+UnMix-TNS) | 49.2 (-40.0) | 43.0 (-44.7) | 49.0 (-36.8) | 41.8 (-45.6) | 45.8 (-41.7) | 49.0 (-40.2) | 41.8 (-45.9) | 47.9 (-37.9) | 41.5 (-45.9) | 45.0 (-42.5) | 49.0 (-44.7) | 42.0 (-51.1) | 47.5 (-42.4) | 39.9 (-53.0) | 44.6 (-47.8) |
| LAME | 32.2 | 29.1 | 28.0 | 32.5 | 30.5 | 32.2 | 29.1 | 28.0 | 32.5 | 30.5 | 21.4 | 10.2 | 11.6 | 17.9 | 15.3 |
| (+UnMix-TNS) | 27.7 (-4.5) | 24.7 (-4.4) | 27.9 (-0.1) | 26.2 (-6.3) | 26.6 (-3.9) | 27.1 (-5.1) | 24.0 (-5.1) | 26.7 (-1.3) | 25.6 (-6.9) | 25.9 (-4.6) | 16.7 (-4.7) | 9.3 (-0.9) | 11.6 (+0.0) | 12.9 (-5.0) | 12.6 (-2.7) |
| NOTE | 61.0 | 57.1 | 59.1 | 54.8 | 58.0 | 60.8 | 56.8 | 58.7 | 54.4 | 57.6 | 63.8 | 60.2 | 60.9 | 57.6 | 60.6 |
| (+UnMix-TNS) | 50.9 (-10.1) | 44.6 (-12.5) | 49.4 (-9.7) | 42.7 (-12.1) | 46.9 (-11.1) | 50.6 (-10.2) | 44.1 (-12.7) | 48.9 (-9.8) | 42.3 (-12.1) | 46.5 (-11.1) | 49.0 (-14.8) | 43.1 (-17.1) | 50.0 (-10.9) | 39.9 (-17.7) | 45.5 (-15.1) |
| RoTTA | 57.5 | 50.9 | 53.8 | 49.7 | 53.0 | 57.1 | 50.7 | 53.4 | 49.1 | 52.6 | 65.6 | 61.2 | 60.2 | 59.6 | 61.7 |
| (+UnMix-TNS) | 50.5 (-7.0) | 44.1 (-6.8) | 49.3 (-4.5) | 43.0 (-6.7) | 46.7 (-6.3) | 50.4 (-6.7) | 43.6 (-7.1) | 49.0 (-4.4) | 43.0 (-6.1) | 46.5 (-6.1) | 49.3 (-16.3) | 42.9 (-18.3) | 49.8 (-10.4) | 40.1 (-19.5) | 45.5 (-16.2) |
| ROID | 88.4 | 86.7 | 85.1 | 86.2 | 86.6 | 88.4 | 86.7 | 85.1 | 86.2 | 86.6 | 93.2 | 92.3 | 89.3 | 92.3 | 91.8 |
| (+UnMix-TNS) | 30.3 (-58.1) | 22.4 (-64.3) | 31.7 (-53.4) | 21.6 (-64.6) | 26.5 (-60.1) | 29.7 (-58.7) | 21.5 (-65.2) | 30.7 (-54.4) | 21.2 (-65.0) | 25.8 (-60.8) | 21.0 (-72.2) | 11.3 (-81.0) | 20.1 (-69.2) | 13.7 (-78.6) | 16.5 (-75.3) |

Table 3: **Non-i.i.d. test-time adaptation on corrupted video datasets.** We adapt the ResNet-50 backbone trained on ImageNet on the sequential frames of ImageNet-VID-C and LaSOT-C and report classification error rates (%). † denotes methods using UnMix-TNS layers.

| Dataset | Corruption | Source | TBN | α-BN | **UnMix-TNS** | TENT | **TENT†** | LAME | **LAME†** |
|---|---|---|---|---|---|---|---|---|---|
| ImageNet-VID-C | Gauss. Noise | 79.6 | 92.4 | 78.1 | **74.1** | 92.4 | 76.3 (-16.1) | 78.0 | 71.2 (-6.8) |
| | Rain | 82.4 | 92.4 | 83.0 | **80.0** | 92.4 | 80.8 (-11.6) | 80.4 | 76.8 (-3.6) |
| | Snow | 49.2 | 91.4 | 61.9 | **51.8** | 91.4 | 54.8 (-36.6) | 45.7 | 46.4 (+0.7) |
| | Mean | 70.4 | 92.0 | 74.4 | **68.6** | 92.0 | 70.6 (-21.4) | 68.0 | **64.8** (-3.2) |
| LaSOT-C | Gauss. Noise | 84.3 | 97.2 | 86.7 | **82.9** | 97.2 | 82.7 (-14.5) | 82.6 | **80.2** (-2.4) |
| | Rain | 89.5 | 97.1 | 89.5 | **88.6** | 97.1 | 88.4 (-8.7) | 87.5 | **87.3** (-0.2) |
| | Snow | 71.2 | 96.1 | 78.4 | **71.9** | 96.1 | 72.1 (-24.0) | 66.5 | 66.8 (+0.3) |
| | Mean | 81.7 | 96.8 | 84.8 | **81.1** | 96.8 | 81.0 (-15.8) | 78.9 | **78.1** (-0.8) |

adaptation, spanning corruption and natural shift benchmarks. More explicitly, our method shows an increase in accuracy by 11.0% and 11.1% averaged over three datasets on the corruption benchmark, surpassing the second-best baseline. The merit is accentuated in DomainNet-126; the improvement rises to 11.5% and 12.5%, corresponding to scenarios where the domain discrepancy is notably profound. In realms of *mixed* domain adaptation, UnMix-TNS outperforms its closest competitor by 5.6% and 2.6% on CIFAR100-C and ImageNet-C, respectively. This is supported by a remarkable reduction in the error rate by 17.2%, surpassing the next best method on *mixed* domain adaptation on DomainNet-126. The insights derived from our experiments are particularly enlightening for non-i.i.d scenarios, suggesting that the update of multiple distinct statistics components achieves superiority over the continuous adaptation of a singular component as in RBN or blending source statistics with the incoming batch/instance ($\alpha$-BN/IABN) target statistics.

**Elevating TTA methods with UnMix-TNS.** In Tables 1 and 2, we benchmark against the forefront TTA strategies, both standalone and when integrated with UnMix-TNS. We note significant performance reductions for TENT, CoTTA, and ROID when operated solely across corruption and natural shift benchmarks, observing error rates escalating to a minimum of 73.4% across every dataset and evaluation scenario. We posit that the reliance of the methods on TBN explains the substantial drops in performance when exposed to non-i.i.d. testing. However, when combined with UnMix-TNS, these methods experience a remarkable performance improvement. Notably, ROID achieves gains of at least 45.8% on corruption datasets and 60.1% on DomainNet-126 domains, ultimately achieving the best mean results in both *single* and *continual* domain adaptation for both benchmarks. Conversely, methods like NOTE and RoTTA use memory banks to simulate i.i.d.-like conditions, improving test-time accuracy in *single* and *continual* domain adaptation scenarios only on corruption benchmarks. Regardless, they face challenges when dealing with *mixed* domain or larger domain shifts, as exemplified by DomainNet-126. Instead, we demonstrate that the integration of UnMix-TNS consistently elevates their performance, highlighting the advanced efficacy of our normalization layers compared to the IABN and RBN employed by these methods. Among the TTA methods, only LAME, which propagates labels within a batch, exhibits remarkably low error rates in both test scenarios. Nonetheless, UnMix-TNS advances its overall performance further, except in the case of *mixed* domain adaptation scenarios for CIFAR100-C and ImageNet-C.

**Evaluating UnMix-TNS for corrupted video datasets.** In Table 3, we assess how a ResNet-50 model, initially trained on ImageNet, adapts at test time using corrupted versions of ImageNet-VID and LaSOT video datasets. UnMix-TNS consistently surpasses the test-time normalization schemes of TBN by 23.4/15.7% and $\alpha$-BN by 5.8/3.7% on ImageNet-VID/LaSOT, respectively. It also demonstrates an overall improvement of 21.4/15.8% and 3.2/0.8% for TENT and LAME, respectively. Moreover, UnMix-TNS proves to be more robust than other test-time normalization schemes, predominantly improving upon the baseline source accuracy, especially in scenarios affected by covariate-shifted video frames, where others may experience a noticeable decline in source model performance.

## 3.3 ABLATION STUDIES

**Deciphering test sample correlation impact.** This study, aligned with (Yuan et al., 2023), adopts the Dirichlet distribution, synthesizing correlatively sampled test streams through the concentration parameter $\delta$ to investigate their impact on domain adaptation under *continual*, and *mixed* domain adaptation. A decrease in $\delta$ results in heightened correlation among test samples and category aggregation, depicted by different $\delta$ values in Figure 3 (a) on the CIFAR100-C dataset. Concurrently, this decrease triggers a pronounced decline in the performance of TBN, $\alpha$-BN, and RBN, owing to their indifference to the rising correlation among test samples. Notably, UnMix-TNS exhibits more resilience regarding various values of $\delta$, demonstrating its effectiveness in the above-mentioned non-i.i.d. test-time scenarios. Further ablation studies concerning *single* domain adaptation, along with experiments on CIFAR10-C, are provided in **Appendix C.**

**Effect of batch size.** While our experiments are primarily conducted with a batch size of 64, we also explore the effect of varying batch sizes, from 1 to 256, on temporally correlated streams within *continual* and *mixed* domain adaptation to address potential curiosities regarding batch size influence. As illustrated in Figure 3 (b), we observe consistent decrement in the error rate for TBN and RBN as batch size increases, while $\alpha$-BN's error rate first increases with batch size and then reduces afterward. This occurrence reflects the premise that an increased batch size facilitates the incorporation of more categories within each batch, thereby diminishing the label correlation therein. For $\alpha$-BN, a batch size of 1 corresponds to the computation of normalization statistics on an instance-wise basis, utilizing a blend of stored and current statistics, a method which, intriguingly, yields the best performance. Distinctly, UnMix-TNS remains robust to batch size variations, consistently delivering superior results across adaptation scenarios over a wide range of batch sizes. Additional results for *single* domain adaptation, along with experiments on CIFAR10-C, are provided in **Appendix C.**

**UnMix-TNS introduces minimal computational overhead.** To accurately assess the efficiency of the UnMix-TNS, we execute precise computations of the inference time over the 15 corruptions of CIFAR10-C. These calculations are uniformly performed under consistent running environments—utilizing an NVIDIA GeForce RTX 3080 GPU and maintaining the same batch size of 64 and $K$=16. Despite the negligible additional inference time cost—a mere extra 0.15ms per image compared to vanilla source inference—the integration of the proposed UnMix-TNS results in a substantial enhancement of 16.5% in average accuracy. When integrated into a method like TENT, the inference rate slightly increases by 0.58ms per image for a significant average accuracy gain of 49.0%.

In **Appendix C**, we provide supplementary ablation studies and experiments, focusing on the effect of the depth of UnMix-TNS layers in neural networks and the number of components, denoted as $K$.

## 4 RELATED WORK

**Online Test-Time Adaptation.** Prominent TTA methods, such as self-supervised tuning (Sun et al., 2020; Liu et al., 2021), batch normalization recalibration (Nado et al., 2020; Wang et al., 2020), and test-time data augmentation (Chen et al., 2022; Tomar et al., 2023), are often limited to specific experimental setups. These methods assume a single stationary distribution shift, large batch sizes, and consistent labels within test batches, leading to suboptimal performance in diverse testing scenarios. Hence, recent efforts have focused on extending to more practical testing scenarios. For instance, CoTTA (Wang et al., 2022) focuses on adapting to evolving target environments but relies on i.i.d. test data. They employ dual utilization of weight and augmentation-averaged predictions,

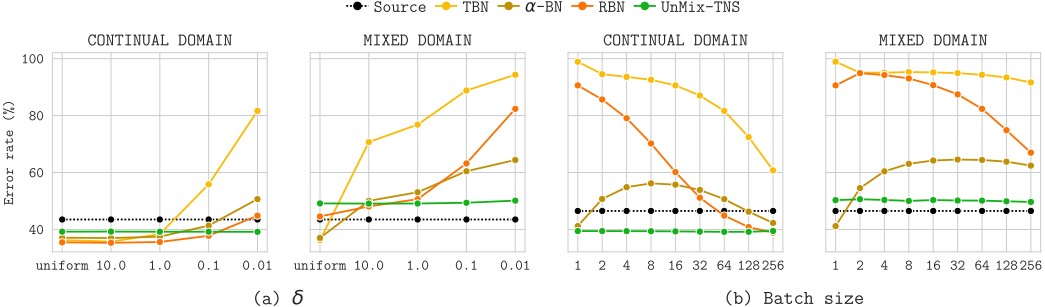

Figure 3: **Ablation study** on the impact of (a) Dirichlet parameter, $\delta$, and (b) batch size on CIFAR100-C, comparing several test-time normalization methods including TBN, $\alpha$-BN, RBN, and UnMix-TNS.

resulting in substantial model updates and computational overhead. LAME (Boudiaf et al., 2022) suggests non-i.i.d. test-time adaptation based on batch predictions, but it may be sensitive to batch size fluctuations. NOTE (Gong et al., 2022) and RoTTA (Yuan et al., 2023) use a memory bank for category-balanced data, effective under non-i.i.d. and non-stationary contexts but with high memory requirements. ROID (Marsden et al., 2024) introduces universal test-time adaptation with various protocols, benefiting from strategies like diversity weighting and normalization layers like group or layer normalization for improved resilience to correlated data. However, its effectiveness diminishes in non-i.i.d. scenarios with BN layers in the backbone.

**Normalization in Test Time.** Test-time BN adaptation methods have emerged that utilize test batch statistics for standardization (Nado et al., 2020) or blending source and test batch statistics (Schneider et al., 2020) to counteract the intermediate covariate shift adeptly. Similarly, methods like $\alpha$-BN (You et al., 2021) and AugBN (Khurana et al., 2021) integrate both statistics through the use of predetermined hyperparameters. Other methods modify statistics via a moving average while augmenting the input to create a virtual test batch (Hu et al., 2021; Mirza et al., 2022). For instance, the MixNorm (Hu et al., 2021) uses training statistics as global statistics, updated through an exponential moving average of online test samples, even for a single sample. InstCal (Zou et al., 2022) introduces an instance-specific BN calibration for test-time adaptation, bypassing extensive test-time parameter fine-tuning. NOTE (Gong et al., 2022) has presented instance-aware BN (IABN) to correct normalization of out-of-distribution samples. In another concurrent work, RoTTA (Yuan et al., 2023) suggests robust BN (RBN), which estimates global statistics via exponential moving average. Recently, TTN (Lim et al., 2023) introduced a test-time normalization layer that merges source and test batch statistics, leveraging interpolating channel-wise weights to seamlessly adapt to new target domains while accounting for domain shift sensitivity. However, it is essential to highlight that TTN relies on prior knowledge of the source data, representing a slight departure from traditional test-time adaptation methods.

## 5 CONCLUSION

This paper proposes UnMix-TNS, a novel test-time normalization layer meticulously designed to counteract the label temporal correlation, particularly in the context of non-i.i.d. distributionally shifted streaming test batches. UnMix-TNS, inherently versatile, integrates seamlessly with existing test-time adaptation techniques and BN-equipped architectures. Through rigorous empirical testing on various benchmarks—including both corruption and natural shift benchmarks of classification, as well as newly introduced corrupted real-world video datasets, we provide compelling evidence of robustness across varied test-time adaptation protocols and the significant performance enhancements achievable by leading TTA methods when paired with UnMix-TNS in non-i.i.d. environments.

**Limitations.** Future work will concentrate on two main areas: firstly, adapting to scenarios where test batches include a wide range of diverse or outlier instances; secondly, applying our method to BN-based segmentation models during test-time adaptation to enhance its adaptability. Additionally, determining the ideal number of UnMix-TNS components will be explored further, as different datasets and adaptation scenarios may require varied optimal values.

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

# A   APPENDIX: PROPERTIES OF UNMIX-TNS COMPONENTS

This section provides supplemental material for Sections 2.2.1 and 2.2.2.

## A.1   STATISTICS OF $K$ COMPONENT MIXTURE DISTRIBUTION

Let $f_X(x) = \sum_k w_k \cdot f_X^k(x)$ be the probability distribution of a random variable $X \in \mathbb{R}$, which is a linear sum of $K$ distinct probability distributions, such that $w_k \geq 0$ and $\sum_k w_k = 1$. Also, let $\mu_k = \int x \cdot f_X^k(x)\, dx$, and $\sigma_k^2 = \int (x - \mu_k)^2 \cdot f_X^k(x)\, dx$. The mean $\mu$ and variance $\sigma^2$ of random variable $X$ under the distribution $f_X(x)$, are computed as:

$$\mu = \int x \cdot f_X(x)\, dx = \int x \cdot \Big( \sum_k w_k \cdot f_X^k(x) \Big)\, dx = \sum_k w_k \int x \cdot f_X^k(x)\, dx = \sum_k w_k \mu_k, \quad (14)$$

$$\sigma^2 = \int (x - \mu)^2 \cdot f_X(x)\, dx = \int (x - \mu)^2 \cdot \Big( \sum_k w_k \cdot f_X^k(x) \Big)\, dx$$

$$= \sum_k w_k \left( \int x^2 \cdot f_X^k(x)\, dx + \mu^2 \int f_X^k(x)\, dx - 2\mu \int x \cdot f_X^k(x)\, dx \right)$$

$$= \sum_k w_k \Big( (\sigma_k^2 + \mu_k^2) + \mu^2 - 2\mu \cdot \mu_k \Big)$$

$$= \sum_k w_k \cdot \sigma_k^2 + \sum_k w_k \cdot \mu_k^2 - \Big( \sum_k w_k \mu_k \Big)^2.$$

## A.2   INITIALIZING INDIVIDUAL STATISTICS OF $K$ UNMIX-TNS COMPONENTS

Let us initialize the means and standard deviations $(\mu_k^t, \sigma_k^t)$ at time $t = 0$ of the individual components of the UnMix-TNS layer, such that, in expectation, the statistics of their mixture distribution are equal to the stored statistics $(\mu, \sigma)$ in the corresponding BN layer. For simplification in notation, we drop $t$ and $c$, and represent the mixture distribution from Section 2.2.1 at $t = 0$ and arbitrary channel $c$ as follows:

$$h_Z(z) = \frac{1}{K} \sum_k \mathcal{N}(\mu_k, \sigma_k) \quad (15)$$

where we define $\mu_k = \mu + \zeta_k$ and $\sigma_k = \rho$, where $\zeta_k \sim \mathcal{N}(0, \epsilon)$. Then, by applying Equations (14) and (15), the expected value $(\bar{\mu}, \bar{\sigma}^2)$ of the mean and variance of mixture distribution can be expressed as follows:

$$\mathbb{E}[\bar{\mu}] = \mathbb{E}\left[ \frac{1}{K} \sum_k (\mu + \zeta_k) \right] = \mu, \quad (16)$$

$$\mathbb{E}[\bar{\sigma}^2] = \mathbb{E}\left[ \rho^2 + \frac{1}{K} \sum_k (\mu + \zeta_k)^2 - \frac{1}{K^2} \Big( \sum_k (\mu + \zeta_k) \Big)^2 \right]$$

$$= \rho^2 + \Big( 1 - \frac{1}{K} \Big) \epsilon^2. \quad (17)$$

Note that the right-hand side of the above equation remains constant. Consequently, at initialization, creating more diverse components (high $\epsilon$) necessitates lower individual variance of the components, $\rho^2$. Thus, the maximum value of $\epsilon^2$ is established as $\epsilon_{\max}^2 = \frac{K}{K-1}\bar{\sigma}^2$ and minimum value as $\epsilon_{\min}^2 = 0$. Hence, we proportionally scale $\epsilon$ between $\Big( 0, \sqrt{\frac{K}{K-1}} \cdot \bar{\sigma} \Big)$ utilizing the hyperparameter $\alpha$, i.e. $\epsilon = \sqrt{\frac{\alpha \cdot K}{K-1}} \cdot \bar{\sigma} \Big)$.

### A.3 ADAPTING UNMIX-TNS TO TEMPORALLY CORRELATED TEST DATA: A THEORETICAL PERSPECTIVE

Let $h(z)$ be the true distribution of the test domain features. We assume that $h(z)$ can be decomposed as a mixture of $K$ Gaussian distributions $\{h_k(z)\}_{k=1}^{K}$, such that $h(z) = \frac{1}{K}\sum_k h_k(z)$. Additionally, we postulate the existence of a perfect classifier $F : Z \to (Y, K)$ that deterministically assigns each feature $z$ to its corresponding label $y$ and component $k$, expressed as $(y, k) = F(z)$.

In the context of independent and identically distributed (i.i.d.) features, each $z$ is uniformly sampled over time with respect to its corresponding label $y$ (unknown to the learner), and thus the expected value of the mean (utilized for normalization) of the current batch can be calculated as follows:

$$\mathbb{E}[\mu_{\text{batch}}] = \mu^* = \sum_{y,k} \int z \cdot h(z, y, k)dz = \sum_{y,k} \int z \cdot h(z|k)h(y|z, k)h(k)dz \tag{18}$$

where $h(z, y, k)$ represents the joint distribution of the features $z$, labels $y$ and the component $k$. The term $h(z|k) = h_k(z)$ denotes the probability distribution of the features given a particular component $k$, while $h(k) = \frac{1}{K}$ implies that each component is equally likely. Furthermore, $h(y|z, k)$ is the conditional probability distribution of the labels given the features $z$ and the component $k$. Since the perfect classifier $F$ allows for the deterministic determination of $y, k$ from $z$, it follows that $\sum_y h(y|z, k) = 1$. This can be rearticulated in the above equation for estimating the expected mean $\mu^*$ as follows:

$$\mu^* = \frac{1}{K}\sum_{y,k} \int z \cdot h_k(z)h(y|z, k)dz = \frac{1}{K}\sum_k \int z \cdot h_k(z)dz = \frac{1}{K}\sum_k \mu_k^*. \tag{19}$$

where $\mu_k^*$ is the mean of the $k^{th}$ component of the true distribution of the test domain features.

In a non-i.i.d. scenario, we sample the features $z$ to ensure a temporal correlation with their corresponding labels $y$. As a result, the true test domain distribution $h(z)$ at a given time $t$ is approximated as $\hat{h}^t(z) = \frac{1}{K}\sum_k \hat{h}_k^t(z)$, where the distribution of the $k^{th}$ component $h_k(z)$ is approximated as $\hat{h}_k^t(z)$. This leads to the estimation of an unbiased normalization mean at time $t$ using $\hat{h}^t(z)$, as expressed in the following equation:

$$\mu_{\text{UnMix-TNS}}(t) = \frac{1}{K}\sum_k \int z \cdot \hat{h}_k^t(z)dz. \tag{20}$$

Furthermore, the bias $\Delta_{\text{UnMix-TNS}}(t)$ between the mean of the true test-domain distribution $\mu^*$ and the estimated mean $\mu_{\text{UnMix-TNS}}(t)$ at time $t$ can be recorded as:

$$\Delta_{\text{UnMix-TNS}}(t) = \frac{1}{K}\sum_k \int z \cdot (h_k(z) - \hat{h}_k^t(z))dz = \frac{1}{K}\sum_k (\mu_k^* - \hat{\mu}_k^t) \tag{21}$$

where $\hat{\mu}_k^t$ is the estimated mean of the same $k^{th}$ component at time $t$. Note that as time progresses and $t$ increases, $\hat{\mu}_k^t \to \mu_k^*$, as we update $\hat{\mu}_k^t$ using exponential moving average mechanism (Equations 12 & 13 in the paper). Thus, UnMix-TNS can help mitigate the bias introduced in the estimation of feature normalization statistics, which arises due to the time-correlated feature distribution. Conversely, the expected value of the normalization mean estimated using the current batch (TBN) for the non-i.i.d. scenario can be defined as:

$$\mu_{\text{TBN}}(t) = \sum_{y,k} \int z \cdot h^t(z, y, k)dz = \sum_{y,k} \int z \cdot h(z|k)h(y|z, k)h^t(k)dz = \sum_k \int z \cdot h_k(z)h^t(k)dz$$

where $h^t(k)$ represents non-i.i.d. characteristics of the test data stream. In this case, the bias is obtained as follows:

$$\Delta_{\text{TBN}}(t) = \mu^* - \sum_k h^t(k)\mu_k^* \tag{22}$$

This equation indicates that if the test samples are uniformly distributed over time (i.e., in an i.i.d. manner), where $h^t(k) = 1/K$, the estimation of normalization statistics will not be biased. However, in situations where $h^t(k)$ smoothly varies with time, favoring the selection of a few components over others, TBN will introduce a non-zero bias in the estimation.

### A.4 Optimizing Momentum: Insights from Momentum Batch Normalization

Our approach to setting the optimal value of the hyperparameter $\lambda$ for momentum draws inspiration from the concept of Momentum BN (MBN) (Yong et al., 2020). The effectiveness of BN has traditionally been attributed to its ability to reduce internal covariate shifts. However, MBN (Yong et al., 2020) has provided an additional perspective by demonstrating that BN layers inherently introduce a certain level of noise in the sample mean and variance, acting as a regularization mechanism during the training phase. A key insight from MBN is the relationship between the amount of noise and the batch size, and smaller batch sizes introduce relatively larger noise, leading to a less stable training process. The rationale behind MBN is to standardize this noise level across different batch sizes, particularly making the noise level with a small batch size comparable to that with a larger batch size by the end of the training stage. To achieve this, MBN modifies the standard BN approach: instead of directly using the batch means and variances in the BN layer, MBN utilizes their momentum equivalents as follows:

$$\mu_c^{t+1} = (1 - \lambda)\mu_c^t + \lambda\mu_B = \mu_c^t + \lambda(\mu_B - \mu_c^t) \tag{23}$$

$$(\sigma_c^{t+1})^2 = (1 - \lambda)(\sigma_c^t)^2 + \lambda(\sigma_B)^2 = (\sigma_c^t)^2 + \lambda((\sigma_B)^2 - (\sigma_c^t)^2) \tag{24}$$

Here, $t$ denotes the $t^{th}$ iteration, $\mu_c^t$ and $\sigma_c^t$ represent historical means and variances, $\mu_B$ and $\sigma_B$ represent the current batch means and variances, and $\lambda$ is the momentum hyperparameter. MBN introduces additive noise $\xi_\mu \sim \mathcal{N}(0, \frac{\lambda}{B})$ and multiplicative noise $\xi_\sigma$ with a Generalized-Chi-squared distribution with expectation $\mathbb{E}[\xi_\sigma] = \frac{B-1}{B}$ and $Var[\xi_\sigma] = \frac{\lambda}{2-\lambda}\frac{2(B-1)}{B^2}$. These formulas show that smaller batch sizes lead to increased noise but also reveal that the noise level can be moderated by the momentum hyperparameter $\lambda$. Based on this insight, MBN proposed a formula to determine a robust $\lambda$ given the batch size $B$:

$$\lambda = 1 - (1 - \lambda_0)^{B/B_0} \tag{25}$$

where $\lambda_0$ and $B_0$ represent the ideal momentum parameter and ideal batch size, respectively. Intuitively, a smaller batch size leads to a lower $\lambda$, thereby reducing noise generation. Given that Equations (12) and (13) in our method are similar to those in the MBN approach Equations (23) and (24), we posit that our method might also encounter significant noise with small batch sizes. To address this and ensure stability across varying batch sizes, we adopt the hyperparameter $\lambda$ following Equation (25), using $B_0 = 64$ and $\lambda_0 = 0.1$, aligning with (Yong et al., 2020).

## B Appendix: For Reproducibility

### B.1 Implementation details

All experiments were performed using PyTorch 1.13 (Paszke et al., 2019) on an NVIDIA GeForce RTX 3080 GPU. For the CIFAR10-C and CIFAR100-C, we optimize the model parameters of the test-time adaptation methods utilizing both BN and UnMix-TNS layers with the Adam optimizer with a learning rate of 1e-5, no weight decay, and a batch size of 64. For the DomainNet-126, ImageNet-C, ImageNet-VID-C, and LaSOT-C datasets, we use the SGD optimizer with a learning rate of 2.5e-6, momentum of 0.9, and no weight decay, with a batch size of 64 for DomainNet-126 and 16 for the remaining datasets.

In implementing our method, we set $\alpha$ to 0.5 in all our experiments. As for the number of UnMix-TNS components $K$, we set 16 for *single* and *continual* test-time adaptation on CIFAR10-C, CIFAR100-C, DomainNet126-C, and ImageNet-C, while setting $K$ to 128 for *mixed* domain test-time adaptation. For *mixed* domain adaptation, $K$ is increased to 128 to aptly represent a diversity of domain features within the neural network and is further adjusted to 256 for ImageNet-VID-C and LaSOT-C. This higher $K$ value is pertinent to accommodate the heterogeneous domain features inherent in *mixed* domain adaptation.

The $\delta$ parameter controlling non-i.i.d. shift of the Dirichlet sampling distribution is set to 0.1 for CIFAR10-C and adjusted to 0.01 for CIFAR100-C, ImageNet-C, and DomainNet-126.

### B.2 PSEUDOCODE

We provide PyTorch-friendly (Paszke et al., 2019) pseudocode for the implementation of the UnMix-TNS layer, referenced as Algorithm 1.

---

**Algorithm 1:** PyTorch-friendly pseudocode for the UnMix-TNS layer

---

```python
class UnMixTNS:
    def __init__(γ, β, source_mean, source_var, momentum, K):
        # initialization
        self.γ = γ, self.β = β, self.momentum = momentum
        # choose α = 0.5, C is number of channels
        α = 0.5, C = source_mean.size()
        # initialize K random UnMix-TNS components
        noise = torch.sqrt(α * K/(K-1)) * torch.randn(K, C) # shape:(K,C)
        self.component_means = torch.tensor([noise[i] * source_mean for i in range(K)]) #
         shape:(K,C)
        self.component_vars = torch.tensor([(1-α) * source_var for _ in range(K)]) #
         shape:(K,C)

    def forward(x):
        # x has shape:(B,C,H,W)
        instance_mean, instance_var = torch.var_mean(x, dim=[2, 3]) # shape:(B,C)
        # compute assignment probabilities
        with torch.no_grad():  # no gradients
            sim = cosine_sim(instance_mean, self.component_means.T) # shape:(B,K)
            p = torch.softmax(sim / 0.07, dim=1).unsqueeze(-1) # shape:(B,K,1)
        # mix the instance statistics with K components' statistics
        hat_mean = (1-p)*self.component_means.unsqueeze(0) + p*instance_mean.unsqueeze(1) #
         shape:  (B,K,C)
        hat_var = (1-p)*self.component_vars.unsqueeze(0) + p*instance_var.unsqueeze(1) #
         shape:  (B,K,C)
        # compute instance-wise normalization statistics
        μ = torch.mean(hat_mean, dim=1)
        σ² = torch.mean(hat_var, dim=1) + torch.mean(hat_mean**2, dim=1) - μ**2
        # update K component's statistics
        with torch.no_grad():  # no gradients
            # update K components' means
            self.component_means = self.component_means + self.momentum *
             (torch.mean(hat_mean, dim=0) - self.component_means)
            # update K components' vars
            self.component_vars = self.component_vars + self.momentum * (torch.mean(hat_var,
             dim=0) - self.component_vars)
        # normalize features :  x
        x_norm = (x - μ) / torch.sqrt(σ² + 1e-6)
        return x_norm * self.γ + self.β
```

---

## C   APPENDIX: ABLATION STUDIES AND ADDITIONAL EXPERIMENTAL RESULTS

### C.1   EXPLORING UNMIX-TNS INFLUENCE ACROSS THE VARIED DEPTHS OF NEURAL NETWORK LAYERS

Figure 4 (a)-(b) illustrate the effect of replacing the BN layer with UnMix-TNS layers in the early or deeper segment of the neural network, respectively. The incorporation of UnMix-TNS proves to be essential for the early and intermediary layers of the neural network for combating the challenges posed by non-i.i.d. test streams. However, in the deeper layers, employing the BN layer synchronized with initial source statistics is discerned to yield superior results.

### C.2   HYPERPARAMETERS SENSITIVITY

In this section, we present supplementary results of our hyperparameter sensitivity analysis, which serves to complement our analysis of the impact of test sample correlation and batch size on the performance of test-time normalization-based methods, UnMix-TNS included. Furthermore, we delve into the sensitivity of UnMix-TNS concerning the number of its components $K$.

**Impact of sample correlation and batch size.**   The resilience of UnMix-TNS, along with other normalization layers, in response to variations in the concentration parameter $\delta$ and batch size, is

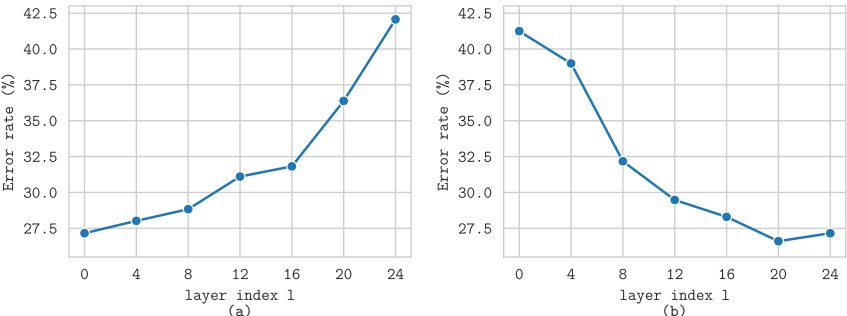

Figure 4: **Exploration of UnMix-TNS influence at varied depths within the neural network.** (a) Represents the average classification error rate when only the BN layers *subsequent* to the *layer index* are replaced by UnMix-TNS layers. (b) Shows the average classification error rate when solely the BN layers *preceding* the *layer index* are exchanged by UnMix-TNS layers. A *layer index* of 0 corresponds to the first layer. The depicted experiments focus on non-i.i.d. *continual* test-time domain adaptation on the CIFAR10-C.

depicted in Figure 5. Furthermore, in Figure 6, we provide additional insights into our sensitivity analysis for CIFAR10-C and CIFAR100-C, showcasing results for *single* test-time domain adaptation. In summary, these additional observations corroborate the analysis in our main paper, underscoring that UnMix-TNS maintains remarkable stability against alterations in batch size and $\delta$, consistently surpassing the average performance of baseline methods.

**Influence of the Number of UNMIX-TNS components $K$.** In Figures 7 and 8, we explore the influence of the hyperparameter $K$ on UnMix-TNS when used both independently and in conjunction with state-of-the-art TTA methods. Our results indicate that for both *single* domain and *continual* domain adaptation, $K = 16$ yields the best performance. However, increasing the value of $K$ results in higher error rates on CIFAR10-C, while the performance remains stable on CIFAR100-C. This discrepancy can presumably be attributed to the limited class range in CIFAR10-C, implicating lesser feature diversification. On the other hand, when it comes to *mixed* domain adaptation, our findings demonstrate that a higher value of $K$ is beneficial for UnMix-TNS. *Mixed* domain adaptation scenarios typically involve more class-wise feature diversity within a batch, which can be effectively handled by increasing the value of $K$.

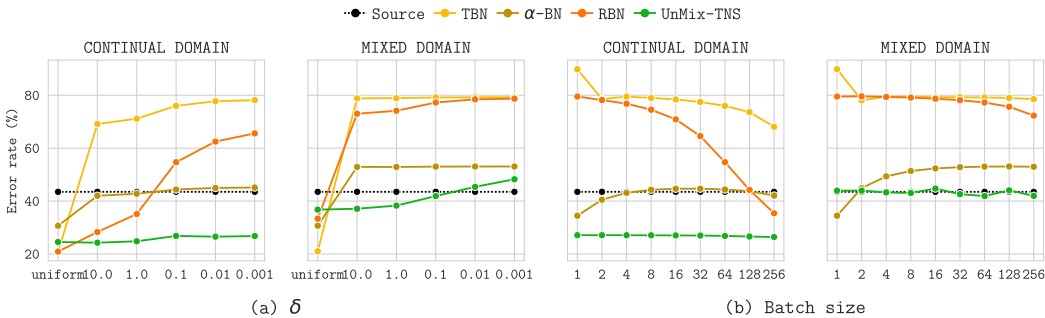

Figure 5: **Ablation study** on the impact of the (a) concentration parameter $\delta$, and (b) batch size on CIFAR10-C for several test-time normalization methods including TBN, $\alpha$-BN, RBN, and our proposed UnMix-TNS.

### C.3 EXPLORATION OF AUGMENTATION TYPES

In the post-training phase, we leverage data augmentation as a means to simulate potential domain shifts, with the primary objective being to subject the model to a spectrum of input domains similar to the corruption benchmark. This simulation is pivotal, allowing an in-depth analysis of the model

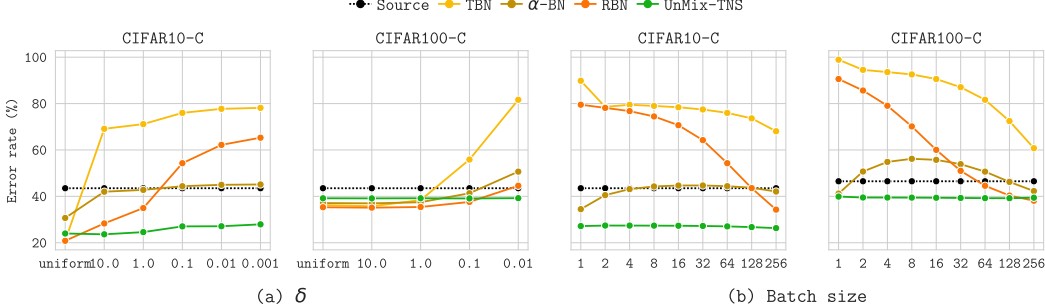

Figure 6: **Ablation study** on the impact of the (a) concentration parameter $\delta$, and (b) batch size on CIFAR10-C and CIFAR100-C for *single* domain adaptation. We compare several test-time normalization methods, including TBN, $\alpha$-BN, RBN, and our proposed UnMix-TNS.

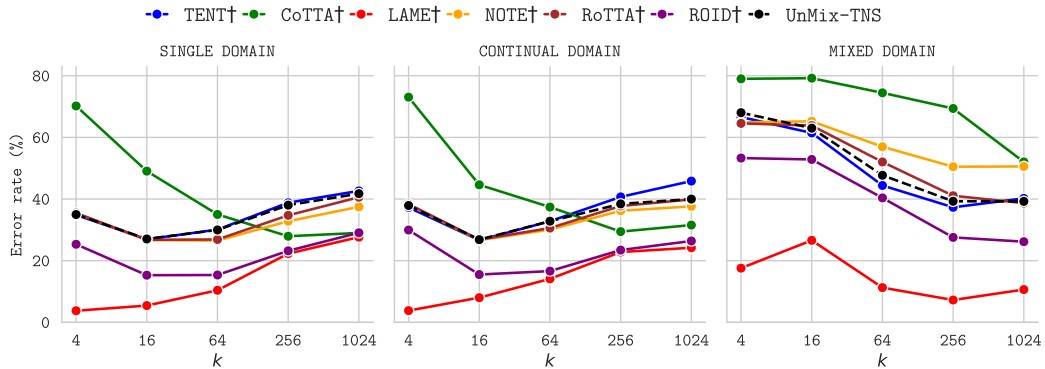

Figure 7: **Ablation study** on the impact of the number of UnMix-TNS components $K$ on CIFAR10-C. The symbol † denotes indicates the employment of UnMix-TNS in the method.

influenced by any discrepancies between shifted and original domains. It is crucial to understand that the essence of this examination is not in the final domain to which the input is altered but in the alteration of the domain itself. This is demonstrated through the ablation study of augmentation types applied on CIFAR10 and CIFAR100 using pre-trained WideResNet-28 and ResNeXt-29 in Table 4.

For this ablation study, color jittering is employed as the base augmentation, and we sequentially incorporated random grayscale, Gaussian blur, and random horizontal flip—each augmentative step reflective, though not identically aligned, with the corruption types present in our corruption benchmark. Our observations reveal that UnMix-TNS consistently maintains a lower average error rate across all augmentations, aligning closely with the source model and even surpassing it in instances such as Gaussian blur. These findings are supported by corresponding non-i.i.d. test-time adaptations on corruption benchmarks, detailed in Table 1, substantiating the versatile efficacy of UnMix-TNS in navigating diverse domain alterations while preserving the accuracy of the model.

## C.4  UNMIX-TNS SAFEGUARDS THE KNOWLEDGE SOURCED FROM THE ORIGINAL DOMAIN

In practical scenarios, data originating from the source domain may resurface during test time. For this purpose, we employed clean domain test data from CIFAR10 and CIFAR100 datasets in a *single* domain non-i.i.d. adaptation scenario to demonstrate how UnMix-TNS and other test-time normalization methods acclimate to previously encountered source domain data, albeit unseen instances. As depicted in Table 5, all baseline strategies employing test-time normalization layers exhibit a decrement in performance, regardless of the expansion in batch sizes. This leads us to infer that relying on source statistics amassed from extensive training data is still preferable to depending solely on current input statistics. However, UnMix-TNS distinguishes itself by continuously refining the

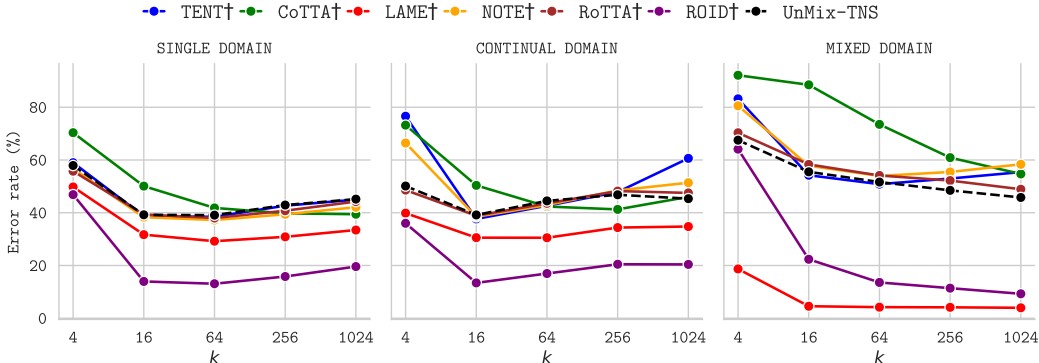

Figure 8: **Ablation study** on the impact of the number of UnMix-TNS components $K$ on CIFAR100-C. The symbol † denotes indicates the employment of UnMix-TNS in the method.

Table 4: **Robustness to data augmentation**. The error rate ($\downarrow$) on augmented CIFAR10 and CIFAR100 using pre-trained WideResNet-28 and ResNeXt-29, respectively. +X denotes the augmentation X is added sequentially to color jittering in the transformation function.

| Dataset | CIFAR10 | | | | CIFAR100 | | | |
|---|---|---|---|---|---|---|---|---|
| Augmentation | color jitter | +grayscale | +gaussian blur | +horizontal flip | color jitter | +grayscale | +gaussian blur | +horizontal flip |
| Source | 7.3 | 11.9 | 55.3 | 7.1 | 27.0 | 53.2 | 49.6 | 27.5 |
| TBN | 72.0 | 73.2 | 76.5 | 71.8 | 80.0 | 84.9 | 82.4 | 79.8 |
| $\alpha$-BN | 18.2 | 23.6 | 56.0 | 17.8 | 40.1 | 57.9 | 55.7 | 40.5 |
| RBN | 43.0 | 47.0 | 60.7 | 43.0 | 38.6 | 57.5 | 50.6 | 39.1 |
| **UnMix-TNS** | **10.8** | **16.0** | **43.0** | **10.7** | **31.4** | **52.1** | **46.2** | **32.1** |

statistics components of the normalization layer, initialized from source statistics, whilst concurrently utilizing the current input. This strategy facilitates enhanced preservation of source knowledge, as demonstrated by a more moderate decline in performance compared to the source model, particularly on the CIFAR100 dataset. Moreover, UnMix-TNS manifests stability in a non-i.i.d. adaptation scenario and exhibits robustness to variations in batch size. This contrasts with other normalization methods, which necessitate larger batch sizes to produce reliable statistics.

Table 5: **Non-i.i.d adaptation in source domain.** The error rate ($\downarrow$) on CIFAR10 and CIFAR100 using WideResNet-28 and ResNeXt-29, respectively.

| Method | Source | TBN | | | $\alpha$-BN | | | RBN | | | **UnMix-TNS** | | |
|---|---|---|---|---|---|---|---|---|---|---|---|---|---|
| Batch size | - | 64 | 16 | 4 | 64 | 16 | 4 | 64 | 16 | 4 | 64 | 16 | 4 |
| CIFAR10 | 5.2 | 70.8 | 74.2 | 76.0 | 15.0 | 15.0 | 13.0 | 40.7 | 63.4 | 71.8 | **8.6** | **8.9** | **8.9** |
| CIFAR100 | 21.1 | 77.9 | 89.0 | 92.2 | 34.2 | 39.3 | 37.7 | 32.7 | 50.3 | 73.9 | **25.4** | **25.6** | **25.7** |
| Avg. | 13.2 | 74.4 | 81.6 | 84.1 | 24.6 | 27.2 | 25.4 | 36.7 | 56.9 | 72.9 | **17.0** | **17.3** | **17.3** |

## C.5 CORRUPTION-SPECIFIC RESULTS

In Tables 6 to 8, we provide a comprehensive set of results for CIFAR10-C, CIFAR100-C, and ImageNet-C, focusing on *single*, *continual*, and *mixed* domain adaptation. More precisely, we delineate error rates for each individual corruption within the benchmark. In the case of *continual* domain adaptation, the corruptions are ordered based on the test timestamps. These results reinforce our prior analysis from the main paper, offering a more granular examination of the corruption level. In particular, for *single* domain adaptation, we consistently outperform test-time normalization-based methods in 14 out of 15 corruption types across all datasets. Within the realm of *continual* domain adaptation, our approach experiences slightly higher error rates on a few corruptions, those of motion, defocus, fog, and contrast with IABN for CIFAR10/100-C. Nonetheless, we sustain a superior stance in terms of overall performance. As for ImageNet-C, UnMix-TNS mirrors the outcomes of RBN in the presence of noisy data but consistently outperforms RBN when subjected to alternative corruptions. These elaborate findings robustly validate the proficiency of UnMix-TNS amidst varying circumstances and corruption types.

Table 6: **Corruption-specific error rates under three non-i.i.d. test-time adaptation scenarios.** The depicted error rates correspond to the adaptation from CIFAR10 to CIFAR10-C on temporally correlated samples by class labels using Dirichlet distribution with $\delta = 0.1$ and corruption severity of 5. Methods marked with [†] denote the integration of our proposed UnMix-TNS with the respective TTA method. Averaged over three runs.

### (a) Single domain non-i.i.d. test-time adaptation.

| METHOD | GAUSS | SHOT | IMPUL. | DEFOC. | GLASS | MOTION | ZOOM | SNOW | FROST | FOG | BRIGHT. | CONTR. | ELAST. | PIXEL | JPEG | AVG. |
|---|---|---|---|---|---|---|---|---|---|---|---|---|---|---|---|---|
| Source | 72.3 | 65.7 | 72.9 | 47.0 | 54.3 | 34.8 | 42.0 | 25.1 | 41.3 | 26.0 | 9.3 | 46.7 | 26.6 | 58.5 | 30.3 | 43.5 |
| TEST TIME NORMALIZATION | | | | | | | | | | | | | | | | |
| TBN | 77.8 | 77.2 | 80.1 | 73.9 | 80.6 | 74.2 | 74.2 | 75.5 | 74.3 | 73.5 | 72.0 | 73.6 | 77.7 | 76.7 | 78.7 | 76.0 |
| $\alpha$-BN | 60.7 | 56.3 | 66.5 | 41.8 | 58.0 | 38.0 | 40.0 | 35.4 | 39.8 | 31.6 | 19.2 | 38.6 | 41.5 | 54.0 | 44.3 | 44.4 |
| IABN | 43.5 | 40.7 | 51.2 | 20.6 | 48.6 | 19.5 | 21.8 | 22.0 | 24.0 | 21.2 | 11.3 | 10.9 | 34.1 | 30.5 | 36.6 | 29.1 |
| RBN | 59.8 | 58.1 | 65.2 | 49.2 | 65.4 | 50.1 | 48.7 | 52.3 | 51.6 | 49.4 | 43.4 | 48.2 | 57.7 | 55.2 | 60.0 | 54.3 |
| **UnMix-TNS** | **38.9** | **35.9** | **46.6** | **20.4** | **41.4** | **19.1** | **20.1** | **21.4** | **22.9** | **19.1** | **11.1** | **16.7** | **30.4** | **27.8** | **33.9** | **27.0** |
| TEST TIME OPTIMIZATION | | | | | | | | | | | | | | | | |
| TENT | 77.8 | 77.2 | 80.1 | 73.8 | 80.6 | 74.2 | 74.2 | 75.5 | 74.3 | 73.4 | 72.0 | 73.5 | 77.7 | 76.7 | 78.8 | 76.0 |
| **TENT**[†] | 38.7 | 35.8 | 46.5 | 20.3 | 41.4 | 19.1 | 20.0 | 21.3 | 22.8 | 19.0 | 11.0 | 16.6 | 30.4 | 27.8 | 33.9 | 27.0 |
| CoTTA | 77.8 | 77.1 | 80.1 | 76.1 | 80.9 | 76.7 | 75.9 | 76.0 | 75.1 | 74.2 | 78.4 | 78.3 | 77.1 | 78.0 | 77.2 |
| **CoTTA**[†] | 68.1 | 63.8 | 72.7 | 37.7 | 73.1 | 35.1 | 38.9 | 43.0 | 42.4 | 35.0 | 16.2 | 34.6 | 60.2 | 52.6 | 62.6 | 49.1 |
| LAME | 77.8 | 67.4 | 63.0 | 25.4 | 42.2 | 12.4 | 14.2 | 9.3 | 26.2 | 11.2 | 4.6 | 39.5 | 5.9 | 53.0 | 6.3 | 30.6 |
| **LAME**[†] | **7.8** | **7.4** | **11.4** | **4.2** | **5.1** | **4.1** | **4.0** | **4.4** | **4.9** | **4.8** | **3.8** | **5.2** | **4.8** | **5.3** | **4.3** | **5.4** |
| NOTE | 37.9 | 35.3 | 44.6 | 18.2 | 44.4 | 17.2 | 19.7 | 20.1 | 21.0 | 18.7 | 10.6 | 9.3 | 31.6 | 27.7 | 34.1 | 26.0 |
| **NOTE**[†] | 37.5 | 35.2 | 46.1 | 19.1 | 42.0 | 18.9 | 18.8 | 21.6 | 22.8 | 19.8 | 11.3 | 16.4 | 30.3 | 26.4 | 34.4 | 26.7 |
| RoTTA | 37.9 | 35.6 | 46.1 | 18.5 | 46.2 | 19.4 | 17.8 | 23.2 | 23.7 | 20.5 | 10.9 | 20.7 | 31.5 | 27.6 | 35.9 | 27.7 |
| **RoTTA**[†] | 37.7 | 35.1 | 46.2 | 19.2 | 42.1 | 18.9 | 18.8 | 21.5 | 23.0 | 19.9 | 11.3 | 16.5 | 30.5 | 26.7 | 34.4 | 26.8 |
| ROID | 75.6 | 74.9 | 78.7 | 70.7 | 79.2 | 71.1 | 71.1 | 72.7 | 71.2 | 70.0 | 68.5 | 70.2 | 75.7 | 74.2 | 76.8 | 73.4 |
| **ROID**[†] | 24.4 | 22.5 | 32.3 | 8.4 | 27.9 | 7.7 | 7.6 | 10.3 | 11.2 | 9.4 | 3.9 | 7.3 | 17.1 | 17.7 | 21.7 | 15.3 |

### (b) Continual domain non-i.i.d. test-time adaptation.

$t \longrightarrow$

| METHOD | GAUSS | SHOT | IMPUL. | DEFOC. | GLASS | MOTION | ZOOM | SNOW | FROST | FOG | BRIGHT. | CONTR. | ELAST. | PIXEL | JPEG | AVG. |
|---|---|---|---|---|---|---|---|---|---|---|---|---|---|---|---|---|
| Source | 72.3 | 65.7 | 72.9 | 47.0 | 54.3 | 34.8 | 42.0 | 25.1 | 41.3 | 26.0 | 9.3 | 46.7 | 26.6 | 58.5 | 30.3 | 43.5 |
| TEST TIME NORMALIZATION | | | | | | | | | | | | | | | | |
| TBN | 77.8 | 77.2 | 80.1 | 73.9 | 80.6 | 74.2 | 74.2 | 75.5 | 74.3 | 73.5 | 72.0 | 73.6 | 77.7 | 76.7 | 78.7 | 76.0 |
| $\alpha$-BN | 60.7 | 56.3 | 66.5 | 41.8 | 58.0 | 38.0 | 40.0 | 35.4 | 39.8 | 31.6 | 19.2 | 38.6 | 41.5 | 54.0 | 44.3 | 44.4 |
| IABN | 43.5 | 40.7 | 51.2 | **20.6** | 48.6 | **19.5** | 21.8 | 22.0 | 24.0 | **21.2** | 11.3 | **10.9** | 34.1 | 30.5 | 36.6 | 29.1 |
| RBN | 59.8 | 57.6 | 64.8 | 51.4 | 65.6 | 50.8 | 48.9 | 52.7 | 50.0 | 44.0 | 48.9 | 58.1 | 55.7 | 60.4 | 54.7 |
| **UnMix-TNS** | **38.9** | **30.8** | **41.1** | 36.0 | **41.0** | 21.4 | **15.3** | **20.1** | **18.8** | 22.2 | **9.9** | 18.4 | **31.2** | **23.8** | **33.5** | **26.8** |
| TEST TIME OPTIMIZATION | | | | | | | | | | | | | | | | |
| TENT | 77.8 | 77.2 | 80.1 | 73.8 | 80.6 | 74.2 | 74.2 | 75.4 | 74.2 | 73.4 | 71.9 | 73.3 | 77.7 | 76.5 | 78.7 | 75.9 |
| **TENT**[†] | 38.7 | 30.5 | 40.9 | 35.8 | 41.2 | 21.2 | 15.1 | 20.2 | 18.6 | 21.7 | 9.9 | 18.2 | 30.8 | 23.5 | 33.1 | 26.6 |
| CoTTA | 77.8 | 77.2 | 80.1 | 76.3 | 81.0 | 77.4 | 76.5 | 77.0 | 76.5 | 76.0 | 75.3 | 79.4 | 79.2 | 78.2 | 79.2 | 77.8 |
| **CoTTA**[†] | 68.1 | 58.8 | 71.6 | 50.8 | 69.3 | 32.4 | 23.8 | 26.5 | 32.9 | 12.6 | 43.0 | 50.8 | 40.9 | 58.8 | 44.6 |
| LAME | 77.8 | 67.4 | 63.0 | 25.4 | 42.2 | 12.4 | 14.2 | 9.3 | 26.2 | 11.2 | 4.6 | 39.5 | 5.9 | 53.0 | 6.3 | 30.6 |
| **LAME**[†] | **7.8** | **4.9** | **6.8** | 14.9 | **7.2** | **6.7** | **6.1** | **8.0** | **8.3** | **8.2** | 5.4 | **9.6** | 10.1 | **7.4** | **8.7** | **8.0** |
| NOTE | 37.9 | 30.8 | 41.6 | 24.1 | 45.1 | 18.4 | 20.1 | 20.2 | 19.7 | 19.3 | 9.4 | 33.0 | 33.2 | 34.3 | 26.7 |
| **NOTE**[†] | 37.5 | 30.4 | 41.9 | 30.7 | 42.4 | 21.1 | 15.5 | 20.5 | 19.7 | 21.4 | 10.5 | 19.6 | 29.5 | 25.7 | 33.4 | 26.7 |
| RoTTA | 37.9 | 33.3 | 44.1 | 25.0 | 46.5 | 20.7 | 16.1 | 22.5 | 22.2 | 21.9 | 10.6 | 20.6 | 33.1 | 27.7 | 35.7 | 27.9 |
| **RoTTA**[†] | 37.7 | 30.4 | 41.7 | 30.0 | 42.8 | 20.9 | 15.9 | 20.8 | 19.7 | 21.6 | 10.8 | 20.8 | 30.7 | 25.3 | 33.4 | 26.8 |
| ROID | 75.6 | 74.8 | 78.6 | 70.7 | 79.2 | 71.1 | 71.1 | 72.7 | 71.2 | 70.0 | 68.5 | 70.2 | 75.7 | 74.2 | 76.8 | 73.4 |
| **ROID**[†] | 24.4 | 18.3 | 28.8 | 15.9 | 30.1 | 8.8 | 6.1 | 10.3 | 9.6 | 9.9 | **4.0** | 10.5 | 16.8 | 18.0 | 21.0 | 15.5 |

### (c) Mixed domain non-i.i.d. test-time adaptation.

| METHOD | GAUSS | SHOT | IMPUL. | DEFOC. | GLASS | MOTION | ZOOM | SNOW | FROST | FOG | BRIGHT. | CONTR. | ELAST. | PIXEL | JPEG | AVG. |
|---|---|---|---|---|---|---|---|---|---|---|---|---|---|---|---|---|
| Source | 72.3 | 65.7 | 72.9 | 47.0 | 54.3 | 34.8 | 42.0 | 25.1 | 41.3 | 26.0 | 9.3 | 46.7 | 26.6 | 58.5 | 30.3 | 43.5 |
| TEST TIME NORMALIZATION | | | | | | | | | | | | | | | | |
| TBN | 90.5 | 88.4 | 97.6 | 80.3 | 90.9 | 77.0 | 77.2 | 70.1 | 70.5 | 73.8 | 43.4 | 74.6 | 84.6 | 85.0 | 83.6 | 79.2 |
| $\alpha$-BN | 76.6 | 71.1 | 85.4 | 57.8 | 64.7 | 45.6 | 52.8 | 33.8 | 45.4 | 35.0 | 14.7 | 56.2 | 42.7 | 67.6 | 46.0 | 53.0 |
| IABN | **43.5** | **40.7** | **51.2** | **20.6** | **48.6** | **19.5** | **21.8** | **22.0** | **24.0** | **21.2** | **11.3** | **10.9** | **34.1** | **30.5** | **36.6** | **29.1** |
| RBN | 88.9 | 86.6 | 96.7 | 78.5 | 89.2 | 74.8 | 75.4 | 67.5 | 68.1 | 71.3 | 41.2 | 73.3 | 82.1 | 84.1 | 81.2 | 77.3 |
| **UnMix-TNS** | 60.4 | 55.9 | 71.5 | 45.9 | 49.6 | 36.2 | 43.2 | 25.0 | 26.8 | 26.2 | 12.6 | 34.4 | 41.1 | 59.7 | 40.1 | 41.9 |
| TEST TIME OPTIMIZATION | | | | | | | | | | | | | | | | |
| TENT | 89.9 | 87.8 | 97.4 | 80.7 | 90.6 | 77.4 | 77.5 | 69.9 | 70.3 | 73.9 | 43.1 | 74.8 | 84.5 | 85.0 | 83.4 | 79.1 |
| **TENT**[†] | 60.3 | 55.0 | 68.7 | 39.3 | 46.1 | 31.3 | 37.0 | 21.6 | 24.6 | 21.9 | 10.9 | 30.8 | 34.7 | 58.5 | 35.0 | 38.4 |
| CoTTA | 91.9 | 89.8 | 96.6 | 82.1 | 90.7 | 80.1 | 78.3 | 73.3 | 73.3 | 81.4 | 50.0 | 90.9 | 83.7 | 80.2 | 78.9 | 81.4 |
| **CoTTA**[†] | 85.6 | 83.1 | 92.0 | 71.4 | 82.4 | 68.1 | 68.4 | 62.2 | 63.8 | 67.8 | 40.2 | 77.9 | 73.5 | 76.1 | 69.4 | 72.2 |
| LAME | 36.1 | 31.5 | 29.9 | 12.4 | 18.7 | 9.8 | 11.6 | 6.9 | 16.2 | 6.8 | 2.5 | 14.4 | 5.8 | 30.6 | 7.7 | 16.1 |
| **LAME**[†] | 15.4 | 13.1 | 16.6 | 6.9 | 6.5 | 5.3 | 7.1 | 2.7 | 3.9 | 3.2 | 1.2 | 6.0 | 5.8 | 20.4 | 5.3 | 8.0 |
| NOTE | 47.8 | 45.5 | 54.7 | 29.0 | 55.7 | 26.3 | 30.9 | 28.8 | 28.0 | 27.8 | 16.5 | 14.3 | 44.4 | 45.8 | 45.3 | 36.1 |
| **NOTE**[†] | 67.8 | 63.7 | 79.9 | 60.5 | 60.9 | 50.6 | 56.2 | 33.9 | 35.8 | 39.6 | 18.5 | 52.3 | 52.8 | 66.1 | 50.2 | 52.6 |
| RoTTA | 76.6 | 73.9 | 88.1 | 67.3 | 77.0 | 63.2 | 64.1 | 49.7 | 49.9 | 56.1 | 26.2 | 61.0 | 68.3 | 74.5 | 63.7 | 64.0 |
| **RoTTA**[†] | 66.0 | 61.8 | 73.8 | 50.1 | 53.1 | 40.9 | 46.4 | 27.1 | 29.5 | 30.2 | 13.6 | 39.9 | 44.7 | 61.5 | 42.1 | 45.4 |
| ROID | 89.3 | 87.0 | 97.1 | 78.2 | 89.8 | 74.8 | 75.1 | 67.2 | 68.0 | 71.7 | 40.2 | 72.6 | 82.7 | 83.7 | 81.4 | 77.3 |
| **ROID**[†] | 53.7 | 49.9 | 61.4 | 29.6 | 40.5 | 24.6 | 28.0 | 18.9 | 22.4 | 19.6 | 8.6 | 24.4 | 28.8 | 49.1 | 30.2 | 32.6 |

Table 7: **Corruption-specific error rates under three non-i.i.d. test-time adaptation scenarios.** The depicted error rates correspond to the adaptation from CIFAR100 to CIFAR100-C on temporally correlated samples by class labels using Dirichlet distribution with $\delta = 0.01$ and corruption severity of 5. Methods marked with [†] denote the integration of our proposed UnMix-TNS with the respective TTA method. Averaged over three runs.

**(a) Single domain non-i.i.d. test-time adaptation.**

| METHOD | GAUSS | SHOT | IMPUL. | DEFOC. | GLASS | MOTION | ZOOM | SNOW | FROST | FOG | BRIGHT. | CONTR. | ELAST. | PIXEL | JPEG | AVG. |
|---|---|---|---|---|---|---|---|---|---|---|---|---|---|---|---|---|
| Source | 73.0 | 68.0 | 39.4 | 29.3 | 54.1 | 30.8 | 28.8 | 39.5 | 45.8 | 50.3 | 29.5 | 55.1 | 37.2 | 74.7 | 41.3 | 46.5 |
| | | | | | | TEST TIME NORMALIZATION | | | | | | | | | | |
| TBN | 83.4 | 82.9 | 83.5 | 79.3 | 83.9 | 79.8 | 79.3 | 81.7 | 81.2 | 83.3 | 79.2 | 80.5 | 82.3 | 80.8 | 83.5 | 81.6 |
| $\alpha$-BN | 64.0 | 61.4 | 50.7 | 40.5 | 57.4 | 42.1 | 40.3 | 48.6 | 49.2 | 56.1 | 38.9 | 49.4 | 48.8 | 59.7 | 52.8 | 50.7 |
| IABN | 65.6 | 64.2 | 65.9 | 47.5 | 66.2 | 47.5 | 47.0 | 52.5 | 53.0 | 64.2 | 43.4 | 38.1 | 59.4 | 57.3 | 63.9 | 55.7 |
| RBN | 51.3 | 49.7 | 51.3 | 36.9 | 50.6 | 38.7 | 37.2 | 44.3 | 43.9 | 51.2 | 35.5 | 39.7 | 45.2 | 42.4 | 50.3 | 44.6 |
| **UnMix-TNS** | **48.3** | **46.1** | **44.0** | **31.5** | **46.6** | **32.0** | **31.0** | **35.9** | **35.9** | **46.9** | **27.6** | **36.4** | **40.5** | **40.5** | **44.9** | **39.2** |
| | | | | | | TEST TIME OPTIMIZATION | | | | | | | | | | |
| TENT | 83.4 | 82.8 | 83.5 | 79.3 | 83.9 | 79.8 | 79.3 | 81.7 | 81.2 | 83.3 | 79.3 | 80.6 | 82.2 | 80.8 | 83.6 | 81.6 |
| **TENT** [†] | 47.7 | 45.5 | 43.2 | 30.9 | 46.2 | 31.4 | 30.6 | 35.7 | 35.6 | 46.3 | 27.4 | 36.0 | 40.0 | 39.6 | 44.3 | 38.7 |
| CoTTA | 81.9 | 81.5 | 82.1 | 79.5 | 82.7 | 79.7 | 79.2 | 81.1 | 80.4 | 82.9 | 78.9 | 80.9 | 81.7 | 79.5 | 81.9 | 80.9 |
| **CoTTA** [†] | 57.4 | 55.8 | 56.9 | 41.8 | 57.3 | 41.9 | 41.0 | 49.5 | 49.2 | 58.8 | 35.4 | 49.0 | 53.3 | 49.0 | 55.0 | 50.1 |
| LAME | 67.3 | 56.9 | 20.0 | 18.4 | 34.3 | 20.0 | 18.7 | 29.4 | 32.8 | 33.7 | 19.1 | 46.1 | 26.6 | 76.3 | 28.7 | 35.2 |
| **LAME** [†] | 36.4 | 35.8 | 33.9 | 28.1 | 34.7 | 27.9 | 28.0 | 31.6 | 30.9 | 34.7 | 24.5 | 28.5 | 33.0 | 33.7 | 33.8 | 31.7 |
| NOTE | 64.1 | 62.0 | 62.4 | 45.2 | 63.0 | 45.6 | 45.1 | 51.4 | 50.9 | 60.8 | 42.6 | 37.7 | 56.7 | 53.8 | 60.3 | 53.4 |
| **NOTE** [†] | 46.3 | 44.5 | 43.3 | 30.4 | 45.4 | 31.3 | 30.4 | 35.4 | 35.5 | 45.7 | 26.8 | 38.6 | 39.5 | 37.6 | 43.9 | 38.3 |
| RoTTA | 51.3 | 50.0 | 50.5 | 33.3 | 48.9 | 35.1 | 32.9 | 41.4 | 45.5 | 46.1 | 31.7 | 52.6 | 43.1 | 40.6 | 48.9 | 43.5 |
| **RoTTA** [†] | 48.0 | 45.9 | 44.6 | 31.2 | 46.4 | 32.2 | 31.3 | 35.8 | 36.1 | 46.4 | 27.1 | 40.0 | 40.5 | 39.3 | 45.4 | 39.4 |
| ROID | 80.4 | 79.7 | 80.5 | 74.3 | 81.1 | 75.0 | 74.2 | 77.8 | 77.0 | 80.4 | 74.2 | 75.7 | 78.6 | 76.6 | 80.4 | 77.7 |
| **ROID** [†] | 21.2 | 18.8 | 16.2 | 8.3 | 18.8 | 8.6 | 8.7 | 11.0 | 10.9 | 18.9 | 6.7 | 12.9 | 13.6 | 16.2 | 18.5 | 14.0 |

**(b) Continual domain non-i.i.d. test-time adaptation.**

$t \longrightarrow$

| METHOD | GAUSS | SHOT | IMPUL. | DEFOC. | GLASS | MOTION | ZOOM | SNOW | FROST | FOG | BRIGHT. | CONTR. | ELAST. | PIXEL | JPEG | AVG. |
|---|---|---|---|---|---|---|---|---|---|---|---|---|---|---|---|---|
| Source | 73.0 | 68.0 | 39.4 | 29.3 | 54.1 | 30.8 | 28.8 | 39.5 | 45.8 | 50.3 | 29.5 | 55.1 | 37.2 | 74.7 | 41.3 | 46.5 |
| | | | | | | TEST TIME NORMALIZATION | | | | | | | | | | |
| TBN | 83.4 | 82.9 | 83.5 | 79.3 | 83.9 | 79.8 | 79.3 | 81.7 | 81.2 | 83.3 | 79.2 | 80.5 | 82.3 | 80.8 | 83.5 | 81.6 |
| $\alpha$-BN | 64.0 | 61.4 | 50.7 | 40.5 | 57.4 | 42.1 | 40.3 | 48.6 | 49.2 | 56.1 | 38.9 | 49.4 | 48.8 | 59.7 | 52.8 | 50.7 |
| IABN | 65.6 | 64.2 | 65.9 | 47.5 | 66.2 | 47.5 | 47.0 | 52.5 | 53.0 | 64.2 | 43.4 | **38.1** | 59.4 | 57.3 | 63.9 | 55.7 |
| RBN | 51.3 | 49.6 | 51.7 | 38.1 | 51.1 | 38.9 | 37.3 | 44.6 | 44.0 | 51.7 | 35.7 | 40.0 | 45.5 | 42.8 | 50.6 | 44.9 |
| **UnMix-TNS** | **48.3** | **43.8** | **43.8** | **34.0** | **47.5** | **32.4** | **29.7** | **34.8** | **35.5** | **46.8** | **27.4** | 38.5 | **43.1** | **38.7** | **43.3** | **39.2** |
| | | | | | | TEST TIME OPTIMIZATION | | | | | | | | | | |
| TENT | 83.4 | 82.7 | 83.4 | 79.6 | 83.8 | 80.1 | 79.5 | 82.0 | 81.5 | 83.6 | 79.8 | 81.4 | 82.7 | 81.4 | 83.9 | 81.9 |
| **TENT** [†] | 47.7 | 41.9 | 41.3 | 32.0 | 46.4 | 30.4 | 28.0 | 34.2 | 34.3 | 44.6 | 27.3 | 36.2 | 41.5 | 37.3 | 41.7 | 37.7 |
| CoTTA | 81.9 | 81.5 | 82.2 | 79.5 | 82.9 | 79.9 | 79.4 | 81.5 | 80.8 | 83.4 | 79.4 | 81.6 | 82.2 | 80.0 | 82.3 | 81.2 |
| **CoTTA** [†] | 57.4 | 56.4 | 57.4 | 41.5 | 57.1 | 40.9 | 38.9 | 49.1 | 49.3 | 59.7 | 35.2 | 53.4 | 54.1 | 49.5 | 55.8 | 50.4 |
| LAME | 67.3 | 56.9 | 20.0 | 18.4 | 34.3 | 20.0 | 18.7 | 29.4 | 32.8 | 33.7 | 19.1 | 46.1 | 26.6 | 76.3 | 28.7 | 35.2 |
| **LAME** [†] | 36.4 | 33.7 | 33.5 | 27.6 | 34.7 | 26.6 | 25.7 | 29.9 | 30.3 | 33.3 | 22.6 | 27.1 | 32.6 | 32.0 | 32.1 | 30.6 |
| NOTE | 64.1 | 58.5 | 59.2 | 48.4 | 62.9 | 46.5 | 45.0 | 52.5 | 49.8 | 59.6 | 44.8 | 40.4 | 57.3 | 57.0 | 61.6 | 53.8 |
| **NOTE** [†] | 46.3 | 41.4 | 41.5 | 31.3 | 45.2 | 30.5 | 28.5 | 35.1 | 36.0 | 45.3 | 28.5 | 40.6 | 41.9 | 39.8 | 44.9 | 38.5 |
| RoTTA | 51.3 | 51.2 | 52.3 | 36.0 | 50.9 | 35.6 | 32.8 | 41.0 | 44.7 | 49.2 | 30.3 | 53.1 | 47.5 | 42.2 | 46.5 | 44.3 |
| **RoTTA** [†] | 48.0 | 44.5 | 44.5 | 33.7 | 46.7 | 31.9 | 29.7 | 35.3 | 36.2 | 44.9 | 26.9 | 40.6 | 40.2 | 37.1 | 43.0 | 38.9 |
| ROID | 80.4 | 79.7 | 80.5 | 74.2 | 81.1 | 75.0 | 74.2 | 77.8 | 76.9 | 80.4 | 74.1 | 75.7 | 78.5 | 76.5 | 80.4 | 77.7 |
| **ROID** [†] | **21.2** | **16.6** | **15.8** | **9.6** | **19.1** | **8.2** | **7.7** | **10.6** | **10.7** | **17.3** | **6.4** | **13.5** | **13.0** | **15.0** | **16.2** | **13.4** |

**(c) Mixed domain non-i.i.d. test-time adaptation.**

$t \longrightarrow$

| METHOD | GAUSS | SHOT | IMPUL. | DEFOC. | GLASS | MOTION | ZOOM | SNOW | FROST | FOG | BRIGHT. | CONTR. | ELAST. | PIXEL | JPEG | AVG. |
|---|---|---|---|---|---|---|---|---|---|---|---|---|---|---|---|---|
| Source | 73.0 | 68.0 | 39.4 | 29.3 | 54.1 | 30.8 | 28.8 | 39.5 | 45.8 | 50.3 | 29.5 | 55.1 | 37.2 | 74.7 | 41.3 | 46.5 |
| | | | | | | TEST TIME NORMALIZATION | | | | | | | | | | |
| TBN | 97.9 | 97.9 | 96.0 | 90.7 | 96.8 | 91.0 | 90.4 | 93.3 | 93.5 | 97.2 | 85.6 | 94.2 | 95.9 | 98.3 | 96.3 | 94.3 |
| $\alpha$-BN | 83.8 | 80.4 | 59.1 | 47.3 | 70.9 | 48.9 | 47.7 | 60.6 | 63.3 | 73.2 | 46.2 | 72.7 | 60.4 | 87.6 | 64.1 | 64.4 |
| IABN | **65.6** | **64.2** | 65.9 | 47.5 | 66.2 | 47.5 | 47.0 | 52.5 | 53.0 | 64.2 | 43.4 | **38.1** | 59.4 | **57.3** | 63.9 | 55.7 |
| RBN | 90.9 | 90.2 | 83.6 | 74.9 | 86.7 | 75.2 | 74.4 | 79.9 | 79.3 | 87.7 | 67.9 | 83.0 | 83.3 | 93.6 | 85.0 | 82.4 |
| **UnMix-TNS** | 70.1 | 67.8 | **44.3** | **39.8** | **53.7** | **39.3** | **38.7** | **42.2** | **43.6** | **53.8** | **32.6** | 49.4 | **45.3** | 79.3 | **51.8** | **50.1** |
| | | | | | | TEST TIME OPTIMIZATION | | | | | | | | | | |
| TENT | 98.1 | 98.0 | 96.6 | 91.3 | 97.0 | 91.6 | 90.9 | 93.6 | 94.2 | 97.5 | 86.0 | 95.5 | 95.9 | 98.5 | 96.3 | 94.7 |
| **TENT** [†] | 75.0 | 72.0 | 50.5 | 34.8 | 58.9 | 35.7 | 34.2 | 42.0 | 46.2 | 53.9 | 32.2 | 54.2 | 42.8 | 86.5 | 49.7 | 51.2 |
| CoTTA | 97.2 | 97.1 | 97.5 | 91.2 | 96.2 | 91.4 | 90.1 | 93.9 | 93.1 | 97.4 | 88.3 | 97.9 | 95.2 | 93.2 | 94.7 | 94.3 |
| **CoTTA** [†] | 81.7 | 80.7 | 71.8 | 56.8 | 71.2 | 57.2 | 54.9 | 62.8 | 63.0 | 72.9 | 50.3 | 74.7 | 63.8 | 71.1 | 65.6 | 66.6 |
| LAME | 4.7 | **4.4** | **3.2** | **3.2** | 3.8 | 3.4 | 3.4 | 3.8 | 3.7 | 3.9 | 3.4 | 3.7 | 3.6 | **4.9** | 3.7 | 3.8 |
| **LAME** [†] | **4.3** | 4.5 | 4.0 | 4.0 | 4.2 | 4.0 | 4.0 | 4.3 | 4.0 | 4.3 | 4.0 | 4.2 | 4.3 | 4.5 | 4.3 | 4.2 |
| NOTE | 68.0 | 66.3 | 65.0 | 50.1 | 65.8 | 50.3 | 49.6 | 54.2 | 52.5 | 63.3 | 46.0 | 40.4 | 58.1 | 62.3 | 63.5 | 57.0 |
| **NOTE** [†] | 73.8 | 71.2 | 51.8 | 39.9 | 57.8 | 40.5 | 39.3 | 44.6 | 48.5 | 58.8 | 35.2 | 65.6 | 46.3 | 82.1 | 51.9 | 53.8 |
| RoTTA | 80.7 | 80.6 | 68.7 | 57.3 | 71.3 | 57.1 | 56.5 | 57.5 | 54.0 | 69.6 | 41.2 | 60.4 | 65.5 | 84.7 | 69.2 | 65.0 |
| **RoTTA** [†] | 72.8 | 71.3 | 49.3 | 44.1 | 56.3 | 43.1 | 42.5 | 44.6 | 45.8 | 56.5 | 33.5 | 52.8 | 48.7 | 81.5 | 55.4 | 53.2 |
| ROID | 97.5 | 97.5 | 95.4 | 89.6 | 96.3 | 90.0 | 89.3 | 92.4 | 92.6 | 96.7 | 84.0 | 93.4 | 95.2 | 97.9 | 95.5 | 93.5 |
| **ROID** [†] | 25.5 | 23.3 | 9.4 | 7.2 | 12.1 | 6.8 | 6.6 | 7.8 | 8.8 | 10.1 | 5.2 | 10.0 | 8.1 | 32.6 | 11.6 | 12.3 |

Table 8: **Corruption-specific error rates under three non-i.i.d. test-time adaptation scenarios.** The depicted error rates correspond to the adaptation from ImageNet to ImageNet-C on temporally correlated samples by class labels using Dirichlet distribution with $\delta = 0.01$ and corruption severity of 5. Methods marked with $\dagger$ denote the integration of our proposed UnMix-TNS with the respective TTA method. Averaged over three runs.

### (a) Single domain non-i.i.d. test-time adaptation.

| METHOD | GAUSS | SHOT | IMPUL. | DEFOC. | GLASS | MOTION | ZOOM | SNOW | FROST | FOG | BRIGH. | CONTR. | ELAST. | PIXEL | JPEG | AVG. |
|---|---|---|---|---|---|---|---|---|---|---|---|---|---|---|---|---|
| Source | 97.8 | 97.1 | 98.2 | 81.7 | 89.8 | 85.2 | 77.9 | 83.5 | 77.1 | 75.9 | 41.3 | 94.5 | 82.5 | 79.3 | 68.6 | 82.0 |
| | | | | | | | TEST TIME NORMALIZATION | | | | | | | | | |
| TBN | 91.9 | 91.5 | 91.7 | 92.6 | 92.3 | 86.4 | 80.2 | 81.3 | 81.8 | 74.0 | 62.8 | 91.1 | 76.7 | 74.4 | 79.0 | 83.2 |
| $\alpha$-BN | 89.3 | 88.7 | 88.2 | 81.6 | 85.7 | 81.2 | 74.1 | 75.4 | 71.5 | 65.6 | 44.0 | 88.3 | 70.1 | 67.5 | 64.8 | 75.7 |
| IABN | 94.8 | 94.8 | 94.3 | 94.0 | 94.3 | 86.5 | 87.3 | 80.9 | 81.2 | 77.0 | 60.1 | 85.5 | 81.2 | 80.0 | 82.9 | 85.0 |
| RBN | 86.3 | **85.7** | **85.8** | 86.5 | 86.2 | 76.4 | 65.1 | 68.6 | 69.6 | 56.0 | 39.3 | **85.2** | 59.5 | 55.2 | 63.7 | 71.3 |
| UnMix-TNS | 87.2 | **85.7** | 86.3 | 85.5 | 85.1 | 74.1 | 64.4 | 65.9 | 68.3 | 54.7 | 36.7 | 89.3 | 58.2 | 53.6 | 63.5 | 70.6 |
| | | | | | | | TEST TIME OPTIMIZATION | | | | | | | | | |
| TENT | 91.4 | 90.9 | 91.2 | 92.1 | 91.9 | 85.8 | 79.6 | 80.8 | 81.3 | 73.2 | 62.8 | 90.5 | 76.1 | 73.7 | 78.2 | 82.6 |
| TENT$^\dagger$ | 86.6 | 85.1 | 85.7 | 84.4 | 84.1 | 72.7 | 63.1 | 64.5 | 67.5 | 52.9 | 36.2 | 89.7 | 57.2 | 51.8 | 61.7 | 69.5 |
| CoTTA | 91.1 | 90.7 | 90.8 | 92.1 | 91.6 | 86.1 | 79.9 | 81.0 | 81.6 | 73.6 | 62.8 | 90.9 | 76.4 | 74.1 | 78.5 | 82.7 |
| CoTTA$^\dagger$ | 89.2 | 87.6 | 88.9 | 87.3 | 86.7 | 73.5 | 63.5 | 65.0 | 67.8 | 53.6 | 36.4 | 88.3 | 57.7 | 52.7 | 62.6 | 70.7 |
| LAME | 98.6 | 97.8 | 98.8 | 77.9 | 88.9 | 83.0 | 73.0 | 81.8 | 72.7 | 72.8 | 30.9 | 93.9 | 82.1 | 75.4 | 61.9 | 79.3 |
| LAME$^\dagger$ | 85.5 | 83.5 | 84.3 | 82.6 | 82.0 | 67.6 | 55.1 | 57.6 | 61.1 | 43.6 | 25.7 | 87.6 | 48.6 | 42.4 | 54.3 | 64.1 |
| NOTE | 92.6 | 92.3 | 91.9 | 93.5 | 93.6 | 84.4 | 82.6 | 77.1 | 78.0 | 71.5 | 56.3 | 82.8 | 75.9 | 73.7 | 79.6 | 81.7 |
| NOTE$^\dagger$ | 87.1 | 85.9 | 86.3 | 85.8 | 85.3 | 74.7 | 64.8 | 66.3 | 69.0 | 54.9 | 37.2 | 89.2 | 58.9 | 54.1 | 64.2 | 70.9 |
| RoTTA | 86.2 | 85.5 | 84.8 | 87.1 | 87.1 | 76.8 | 63.3 | 67.2 | 69.2 | 52.9 | 35.6 | **86.7** | 57.0 | 52.1 | 61.0 | 70.2 |
| RoTTA$^\dagger$ | 86.9 | 85.5 | 86.2 | 85.9 | 85.4 | 74.5 | 64.9 | 66.1 | 68.8 | 54.5 | 37.0 | 90.1 | 58.6 | 53.9 | 63.6 | 70.8 |
| ROID | 91.5 | 91.1 | 91.3 | 92.2 | 92.0 | 85.9 | 79.5 | 80.5 | 81.1 | 72.6 | 61.1 | 90.5 | 75.7 | 73.2 | 78.0 | 82.4 |
| ROID$^\dagger$ | 87.2 | 85.5 | 86.5 | 83.8 | 83.9 | 70.7 | 59.3 | 60.3 | 63.9 | 45.8 | 26.5 | 88.7 | 52.6 | 46.4 | 57.1 | 66.5 |

### (b) Continual domain non-i.i.d. test-time adaptation.

$t \longrightarrow$

| METHOD | GAUSS | SHOT | IMPUL. | DEFOC. | GLASS | MOTION | ZOOM | SNOW | FROST | FOG | BRIGH. | CONTR. | ELAST. | PIXEL | JPEG | AVG. |
|---|---|---|---|---|---|---|---|---|---|---|---|---|---|---|---|---|
| Source | 97.8 | 97.1 | 98.2 | 81.7 | 89.8 | 85.2 | 77.9 | 83.5 | 77.1 | 75.9 | 41.3 | 94.5 | 82.5 | 79.3 | 68.6 | 82.0 |
| | | | | | | | TEST TIME NORMALIZATION | | | | | | | | | |
| TBN | 91.9 | 91.5 | 91.7 | 92.6 | 92.3 | 86.4 | 80.2 | 81.3 | 81.8 | 74.0 | 62.8 | 91.1 | 76.7 | 74.4 | 79.0 | 83.2 |
| $\alpha$-BN | 89.3 | 88.7 | 88.2 | 81.6 | 85.7 | 81.2 | 74.1 | 75.4 | 71.5 | 65.6 | 44.0 | 88.3 | 70.1 | 67.5 | 64.8 | 75.7 |
| IABN | 94.8 | 94.8 | 94.3 | 94.0 | 94.3 | 86.5 | 87.3 | 80.9 | 81.2 | 77.0 | 60.1 | 85.5 | 81.2 | 80.0 | 82.9 | 85.0 |
| RBN | **86.3** | 85.7 | **85.7** | 86.7 | 86.2 | 76.4 | 65.1 | 68.6 | 69.6 | 56.1 | 39.3 | **85.2** | 59.5 | 55.2 | **63.7** | 71.3 |
| UnMix-TNS | 87.2 | 85.4 | 85.7 | 85.6 | 84.7 | 73.6 | 64.2 | 65.7 | 68.1 | 54.4 | 36.6 | 88.9 | 58.3 | 53.9 | 63.7 | 70.4 |
| | | | | | | | TEST TIME OPTIMIZATION | | | | | | | | | |
| TENT | 91.4 | 90.0 | 89.6 | 90.8 | 90.0 | 84.0 | 78.4 | 80.2 | 80.4 | 73.8 | 65.4 | 88.3 | 75.9 | 74.3 | 77.7 | 82.0 |
| TENT$^\dagger$ | 86.6 | 84.9 | 86.4 | 87.2 | 87.7 | 82.3 | 81.3 | 84.7 | 88.4 | 87.6 | 80.2 | 98.4 | 94.2 | 95.5 | 97.3 | 88.2 |
| CoTTA | 91.1 | 90.5 | 90.6 | 92.0 | 91.4 | 85.8 | 79.6 | 80.9 | 81.3 | 73.3 | 62.8 | 90.5 | 76.3 | 73.9 | 78.2 | 82.6 |
| CoTTA$^\dagger$ | 89.2 | 87.3 | 88.5 | 88.5 | 88.1 | 78.5 | 66.7 | 64.8 | 67.0 | 53.2 | 36.4 | 90.0 | 57.5 | 52.6 | 62.1 | 71.4 |
| LAME | 98.6 | 97.8 | 98.8 | 77.9 | 88.9 | 83.0 | 73.0 | 81.8 | 72.7 | 72.8 | 30.9 | 93.9 | 82.1 | 75.4 | 61.9 | 79.3 |
| LAME$^\dagger$ | 85.5 | 83.2 | 83.6 | 82.6 | 81.5 | 66.8 | 54.7 | 57.3 | 60.5 | 43.4 | 25.7 | 87.0 | 48.5 | 42.4 | 54.4 | 63.8 |
| NOTE | 92.6 | 92.3 | 92.0 | 93.7 | 93.7 | 84.6 | 82.6 | 77.2 | 78.1 | 71.6 | 56.6 | 82.7 | 76.2 | 73.8 | 79.7 | 81.8 |
| NOTE$^\dagger$ | 87.1 | 85.6 | 86.0 | 85.6 | 85.4 | 74.6 | 64.9 | 66.2 | 68.2 | 55.1 | 37.2 | 88.0 | 58.8 | 54.5 | 64.2 | 70.8 |
| RoTTA | 86.2 | 85.5 | 84.3 | 85.8 | 87.5 | 75.6 | 62.2 | 66.5 | 67.6 | 51.8 | 35.0 | **80.9** | 55.5 | 50.9 | 58.1 | 68.9 |
| RoTTA$^\dagger$ | 86.9 | 85.5 | 85.3 | 85.2 | 85.0 | 73.6 | 64.3 | 65.4 | 68.1 | 53.4 | 36.4 | 82.5 | 57.7 | 53.5 | 61.5 | 69.6 |
| ROID | 91.5 | 91.1 | 91.3 | 92.2 | 92.0 | 85.8 | 79.4 | 80.5 | 81.1 | 72.6 | 61.1 | 90.5 | 75.7 | 73.2 | 78.0 | 82.4 |
| ROID$^\dagger$ | 87.2 | 85.1 | 85.7 | 83.8 | 84.1 | 71.0 | 59.4 | 60.2 | 63.7 | 45.7 | 26.6 | 87.2 | 52.6 | 45.9 | 57.4 | 66.4 |

### (c) Mixed domain non-i.i.d. test-time adaptation.

| METHOD | GAUSS | SHOT | IMPUL. | DEFOC. | GLASS | MOTION | ZOOM | SNOW | FROST | FOG | BRIGH. | CONTR. | ELAST. | PIXEL | JPEG | AVG. |
|---|---|---|---|---|---|---|---|---|---|---|---|---|---|---|---|---|
| Source | 97.8 | 97.1 | 98.2 | 81.7 | 89.8 | 85.2 | 77.9 | 83.5 | 77.1 | 75.9 | 41.3 | 94.5 | 82.5 | 79.3 | 68.6 | 82.0 |
| | | | | | | | TEST TIME NORMALIZATION | | | | | | | | | |
| TBN | 99.4 | 99.1 | 99.3 | 98.6 | 99.0 | 98.2 | 97.1 | 96.9 | 96.5 | 96.3 | 81.9 | 99.2 | 97.3 | 95.4 | 94.2 | 96.6 |
| $\alpha$-BN | 96.7 | 96.2 | 96.4 | **89.3** | 93.8 | 91.2 | 85.1 | 88.3 | 84.9 | 82.7 | 52.5 | 97.0 | 89.6 | 81.7 | 75.4 | 86.7 |
| IABN | **94.8** | **94.8** | **94.3** | 94.0 | 94.3 | **86.5** | 87.3 | **80.9** | **81.2** | **77.0** | 60.1 | **85.5** | **81.2** | **80.0** | 82.9 | 85.0 |
| RBN | 97.8 | 96.9 | 97.6 | 95.3 | 96.0 | 94.9 | 91.5 | 91.2 | 90.3 | 89.6 | 64.3 | 97.4 | 92.5 | 87.1 | 84.0 | 91.1 |
| UnMix-TNS | 96.7 | 95.6 | 96.4 | 90.6 | **90.5** | 87.1 | **81.3** | 82.3 | 82.2 | 78.0 | **48.3** | 94.2 | 83.4 | 83.0 | **75.0** | 84.3 |
| | | | | | | | TEST TIME OPTIMIZATION | | | | | | | | | |
| TENT | 99.8 | 99.6 | 99.8 | 99.2 | 99.4 | 99.0 | 97.9 | 97.9 | 97.6 | 98.0 | 82.2 | 99.7 | 98.5 | 98.2 | 97.6 | 97.6 |
| TENT$^\dagger$ | 99.8 | 99.7 | 99.8 | 97.6 | 96.6 | 95.3 | 92.8 | 94.3 | 94.0 | 91.6 | 77.3 | 99.2 | 94.5 | 97.0 | 95.8 | 95.0 |
| CoTTA | 99.6 | 99.4 | 99.5 | 98.8 | 98.9 | 98.3 | 97.1 | 96.9 | 96.6 | 96.4 | 81.4 | 99.3 | 97.3 | 95.9 | 94.7 | 96.7 |
| CoTTA$^\dagger$ | 99.6 | 99.3 | 99.6 | 91.2 | 90.7 | 87.2 | 81.4 | 82.2 | 81.9 | 78.5 | 47.7 | 95.3 | 83.3 | 86.7 | 78.6 | 85.6 |
| LAME | 85.9 | 85.4 | 86.9 | 58.6 | 70.0 | 65.5 | 58.9 | 65.7 | 59.2 | 57.8 | 29.0 | 74.5 | 66.6 | 60.8 | 50.3 | 65.0 |
| LAME$^\dagger$ | 85.1 | 83.4 | 84.4 | 73.2 | 72.8 | 69.3 | 63.3 | **64.2** | 65.1 | 60.3 | 35.3 | 80.5 | **66.2** | 67.3 | 57.5 | 68.5 |
| NOTE | 93.2 | 93.8 | 92.5 | 95.7 | 94.7 | 88.3 | 89.5 | 81.6 | 81.5 | 76.0 | 65.4 | 85.0 | 82.0 | 81.0 | 83.8 | 85.6 |
| NOTE$^\dagger$ | 96.9 | 95.9 | 96.7 | 90.9 | 90.7 | 87.4 | 81.6 | 82.8 | 82.5 | 78.4 | 49.0 | 94.3 | 83.8 | 83.2 | 75.7 | 84.6 |
| RoTTA | 94.8 | 94.3 | 94.7 | 92.5 | 93.9 | 93.2 | 88.2 | 88.2 | 86.1 | 87.0 | 50.6 | 96.6 | 88.3 | 79.5 | 72.3 | 86.7 |
| RoTTA$^\dagger$ | 96.8 | 96.1 | 96.5 | 90.3 | 90.5 | 87.2 | 81.7 | 82.4 | 81.9 | 78.1 | 44.7 | 94.8 | 83.0 | 81.3 | 72.7 | 83.9 |
| ROID | 99.4 | 99.1 | 99.3 | 98.6 | 98.9 | 98.3 | 97.0 | 96.8 | 96.5 | 96.2 | 81.4 | 99.2 | 97.2 | 95.5 | 94.2 | 96.5 |
| ROID$^\dagger$ | 92.5 | 91.0 | 91.8 | 83.2 | 84.1 | 78.8 | 71.9 | 75.3 | 74.9 | 68.4 | 40.6 | 87.4 | 76.0 | 77.9 | 69.3 | 77.5 |

