# OpenReview forum: "Un-Mixing Test-Time Normalization Statistics: Combatting Label Temporal Correlation"
_ICLR.cc/2024/Conference — ICLR 2024 poster_

### Official Review · Reviewer_omE5 · 2023-10-30

**Soundness:** 3 good
**Presentation:** 3 good
**Contribution:** 3 good
**Rating:** 6
**Confidence:** 3

**Summary:**

This paper proposes a novel test-time normalization layer to tackle the temporally correlated, distributionally shifted problems within the context of test-time adaptation, and boost the performance of test-time adaptation tasks involving various distribution shifts. The experiments demonstrate the method achieved SOTA.

**Strengths:**

1. The logic of this paper is clear and the performance is excellent.
The author argues that the assumption of test batches conforming to an i.i.d distribution can produce unstable and unreliable adaptations. This paper works on it and offers a reasonable approach.
2. Extensive experiments on three benchmarks and solid ablation studies.
3. The paper is well organized and the idea is well presented.

**Weaknesses:**

After reading the paper, it is clear that the method delivers improvements on TTA tasks. However, there are some questions remain unanswered:
0. Some papers also explored the test-time non-iid setting, such as [a] and [b]. Compared with these papers, what's the contribution of the proposed setting and approach?
1. Why the hyper-parameter \alpha is set to be 0.5 in Equation (5) for implementation while initializing UNMIX-TNS components?
2. In Section 2.2.3, the hyper-parameter \lambda is not explained clearly.
3. While Tables 1 and 2 include 11 methods for comparison, the experiments on the video dataset (Table 3) only include seven methods.
4. Does the UNMIX-TNS update the same parameters with other TTA methods? And have the affine parameters in BN layers been changed within the optimization? Does the method require only single forward inference or multiple forward passes?
5. The authors mentioned that “only components that closely align with the instance-wise statistics undergo updates”, but equations (12) and (13) update all K BN statistics components with different assignment probabilities.


[a] Robust Test-Time Adaptation in Dynamic Scenarios, ICCV2023
[b] Robust continual test-time adaptation: Instance-aware BN and prediction-balanced memory. NeurIPS2022

**Questions:**

Please refer to the weakness.

---

> ### Author Response · Authors · 2023-11-20
> **Thanks for your comment!**
>
> We appreciate your constructive comments and suggestions, and they are exceedingly helpful for us in improving our paper. We have carefully incorporated them in the revised paper. In the following, your comments are first stated and then followed by our point-by-point responses.
>
> > Comparison w.r.t ([a] RoTTA, [b] NOTE).
>
> Thank you for your comment. In the Introduction, we reference studies ([a] RoTTA, [b] NOTE) that address adaptation under non-i.i.d. data streams. NOTE introduces instance-aware BN (IABN), and RoTTA proposes a robust batch normalization (RBN) module, which uses global statistics to robustly normalize feature maps. However, as our results show, these normalization layers alone struggle in non-i.i.d test-time adaptation scenarios. Their effectiveness is partly due to their use of a memory bank for sampling category-balanced data, thereby simulating an i.i.d. test-time adaptation environment. In contrast, UnMix-TNS's core concept is to maintain both current and past statistics of online test features across its K different components. For each test sample, we refine the most closely aligned components, allowing for the estimation of unbiased BN statistics without relying on a memory bank. Our experiments demonstrate that UnMix-TNS outperforms RBN and IABN in non-i.i.d setups, even when integrated with RoTTA and NOTE, as shown in Tables 1 and 2. This highlights the unique contribution and robustness of our approach in these challenging adaptation scenarios.
>
> > Hyperparameter $\alpha$.
>
> Thank you for your query regarding the choice of setting the hyperparameter $\alpha$. As mentioned in Appendix A.2 of our manuscript, we analyze the source variance within the context of UnMix-TNS. Our analysis indicates that the source variance is contributed by (1) the average variance of individual components, and (2) the variance of the means of individual components. By setting $\alpha=0.5$, we ensure that these two contributions to the source variance are balanced and have equal weight. This balance is crucial for the effective functioning of UnMix-TNS.
>
> > Hyperparameter $\lambda$.
>
> Thank you for highlighting the need for a clearer explanation regarding the hyperparameter $\lambda$ in Section 2.2.3 of our paper. Equations (12) and (13) are derived based on the principles of momentum batch normalization (MBN), and the hyperparameter $\lambda$ is set according to the MBN guidelines proposed by Yong et al. (2020). The primary objective of MBN is to ensure a consistent noise level introduced by BN across varying batch sizes. This consistency is crucial to maintain equivalent noise levels in both small and large batch scenarios. The tuning of $\lambda$ is integral to achieving this standardization based on the specific batch size employed. For a more comprehensive understanding of $\lambda$ and its significance in our method, we have included an in-depth explanation in the newly added Appendix A.4 of the manuscript.
>
> > Method comparison: 11 in Tables 1 and 2, only 7 in video dataset (Table 3).
>
> Thank you for your comment regarding the number of methods included in the comparisons for the video dataset (Table 3). We aimed for a fair and relevant evaluation by aligning our experimental setup with that of LAME (Boudiaf et al., 2022), selecting top-performing baselines for comparison. The methods listed in Tables 1 and 2, which are not included in Table 3, either necessitate extensive reimplementation for application to video datasets or are not directly applicable to such datasets. This decision was made to ensure the most meaningful and practical comparison within the constraints of our experimental setup.
>
> > Query on UNMIX-TNS: parameter updates, BN layer changes, and forward pass requirements.
>
> In response to the reviewer's queries about the UnMix-TNS method, as a standalone method, UnMix-TNS is designed to update only its internal statistics and requires only a single forward pass. However, when integrated with other TTA methods, UnMix-TNS solely replaces the Batch Normalization (BN) layers and behaves like the replaced BN in the same manner implemented in those TTA methods, including parameter updates. For instance,  in a TTA method like TENT, the affine parameters of the BN layers are updated, and UnMix-TNS will follow this same update protocol. Conversely, in methods like LAME, the focus is on updating only the internal statistics of normalization layers. In summary, UnMix-TNS can either operate independently with a single forward pass or align with the update mechanisms of other TTA methods it is paired with.
>
> > Discrepancy in author's claim and Equations (12) \& (13) on updating BN statistics in UNMIX-TNS.
>
> We acknowledge that our initial phrasing may have been misleading. In practice, we update all K BN statistics components, each with varying assignment probabilities. We have updated the manuscript to clarify this point more accurately.

---

> > ### Comment · Reviewer_omE5 · 2023-11-21
> > **Thank you for your response**
> >
> > Thank you for your detailed answers to the other reviewers and me. I keep my originally positive score and look forward to other reviewers' comments.

---

> > > ### Author Response · Authors · 2023-11-23
> > > **Thanks!**
> > >
> > > Dear Reviewer omE5,
> > >
> > > We are glad that we have addressed all of your comments. We hope that our efforts and the additional insights provided will positively influence your final rating, showcasing our dedication to furthering research in the field of Test-Time Adaptation.

---

### Official Review · Reviewer_yuas · 2023-10-31

**Soundness:** 2 fair
**Presentation:** 3 good
**Contribution:** 3 good
**Rating:** 6
**Confidence:** 3

**Summary:**

This paper introduces UnMix-TNS, a new approach to address the challenges of test-time adaptation in neural networks under non-i.i.d. conditions, particularly focusing on data streams with temporal correlation. Traditional batch normalization (BN) techniques, fundamental in stabilizing neural network training, fall short under non-i.i.d. test scenarios, often encountered in real-world applications like autonomous driving. UnMix-TNS innovatively recalibrates instance-wise statistics within BN by mixing them with multiple unmixed components, simulating an i.i.d. environment. This method is adaptable to various state-of-the-art test-time adaptation strategies and seamlessly integrates with pre-trained architectures that include BN layers. The approach is empirically validated across diverse scenarios and shows promise in effectively handling data with temporal correlations. New datasets have also been introduced.

**Strengths:**

1. **Innovative Approach to Test-Time Adaptation:** The introduction of UnMix-TNS recalibrates instance-wise statistics by mixing with multiple unmixed statistics components. It targets the challenge of non-i.i.d. test batch distributions, a limitation in current test-time adaptation methods.
2. **Seamless Integration with Pre-Trained Models:** The ability of UnMix-TNS to integrate with existing pre-trained neural network architectures without significant modifications is also an advantage. This facilitates easier adoption in various applications.
3. **Empirical Validation:** The paper provides empirical evidence of performance improvements across multiple benchmarks, including those with common corruption shifts and natural shifts. UnMix-TNS demonstrates robustness in diverse scenarios. The introduction of new datasets like ImageNet-VID-C and LaSOT-C for realistic domain shifts in frame-wise video classification adds value.

**Weaknesses:**

1. **Lack of Theoretical Insight for Temporal Correlation Handling:** The abstract and introduction do not provide clear theoretical insights or explanations on how UnMix-TNS effectively deals with test data streams having temporal correlation. A deeper understanding of the underlying principles is crucial for assessing the robustness and reliability of the method.
2. **Potential for Bias in Instance Selection:** The effectiveness of UnMix-TNS might be influenced by the selection of instances from incoming test batches for updating statistics. If the selection is not well-designed, it could introduce bias, affecting the generalization of the model.

**Questions:**

- Could the authors provide more detailed theoretical insights or foundational principles on how UnMix-TNS adapts to test data streams with temporal correlation? Specifically, how does the method theoretically account for and mitigate issues that arise due to temporal dependencies in the data?
- What is the theoretical rationale behind the 'unmixing' approach in handling temporally correlated data? How does this strategy compare with traditional methods in terms of theoretical robustness against temporal variations in data streams?
- Could the authors elaborate on the criteria or algorithm used to select instances from incoming test batches for updating the UnMix-TNS statistics? How do the authors ensure that this selection process does not introduce bias, which might affect the model’s generalization capability?
- In situations where the test batches contain highly diverse or outlier instances, how does UnMix-TNS maintain the integrity of the normalization statistics? Is there a mechanism in place to detect and appropriately handle such anomalies?

---

> ### Author Response · Authors · 2023-11-20
> **Thanks for your comment!**
>
> Thank you for your constructive comments and suggestions; and they are exceedingly helpful for us in improving our paper. We have carefully incorporated them in the revised paper. In the following, your comments are first stated and then followed by our point-by-point responses.
>
> > Ambiguity regarding instance selection and potential bias.
>
> We would like to clarify any confusion in the submitted paper regarding the potential for bias in instance selection. For a given batch of test feature-map instances, represented as $\mathbf{z}\in \mathbb{R}^{B\times C\times H \times W}$ (where $B$ is the number of instances, $C$ the number of channels, $H, W$ the width and height of an instance feature-map), **we do not perform any selection of the test features for updating the components statistics**. Instead, every individual feature map $z[b]$ $\in \mathbb{R}^{C\times H \times W}$ is soft-assigned to the $K$ components of UnMix-TNS. This assignment is based on the cosine similarity between the mean of each instance-wise feature map and the centers of the components (Equation 7). Based on this similarity given by $p_{b,k}$, we update the parameters $\mu_k^t$, $\sigma_k^t$ of all $K$ components (Equations 12 \& 13 in the paper). We acknowledge the importance of this clarification.
>
> > More detailed theoretical insights to handling temporally correlated data, comparison with traditional methods and potential bias issue.
>
> We appreciate your thoughtful review and valuable suggestion to provide a deeper theoretical understanding of how our model accounts for temporal correlation. In response, we have expanded our theoretical analysis. In our revised manuscript, we have added a dedicated section (A.3 in the Appendix) to address this topic comprehensively. We also compare UnMix-TNS with the standard test-time normalization method (TBN) in section A.3. Furthermore, we have highlighted why UnMix-TNS can help mitigate the bias introduced in the estimation of feature normalization statistics, which arises due to the time-correlated feature distribution.
>
> >  Dealing with highly diverse or outlier instances.
>
> We appreciate the reviewer's concern about handling highly diverse or outlier instances in test batches. We want to emphasize that we adopt the same setting as other test-time normalization methods for a fair comparison by assuming that temporally correlated test batches are free from outliers, such as instances from novel or unknown categories. This is an important consideration, as the presence of outliers may result in inaccurate normalization statistics estimations in UnMix-TNS, an issue that is also applicable to other methods.
>
> Fortunately, we are already planning to extend our work to include detecting and excluding anomalies during the test-time adaptation process. The envisioned approach involves leveraging an anomaly detection score, calculated based on the distance metric of test instances to the nearest component in the UnMix-TNS and will utilize an optimal threshold to identify outliers effectively. By identifying and pruning outlier instances using this strategy, we aim to prevent the adverse impact these instances might have on adapting the model. This will significantly improve the robustness and accuracy of UnMix-TNS in dealing with diverse test batches. We have outlined this planned improvement as a key focus for our future work in the revised version of our manuscript (conclusion section).

---

> > ### Comment · Reviewer_yuas · 2023-11-21
> > **Thank you for your comments**
> >
> > My concerns have been addressed. Thank you very much.
> >
> > Kind regards,
> >
> > Reviewer yuas

---

> > > ### Author Response · Authors · 2023-11-23
> > > **Thanks!**
> > >
> > > Dear Reviewer yuas,
> > >
> > > We are pleased that we could address all your comments, especially the newly added section on theoretical analysis for label temporal correlation. We hope this additional insight and our efforts in this direction positively impact your final rating and demonstrate our commitment to advancing the field of Test-Time Adaptation.

---

### Official Review · Reviewer_auPw · 2023-11-01

**Soundness:** 3 good
**Presentation:** 2 fair
**Contribution:** 3 good
**Rating:** 8
**Confidence:** 3

**Summary:**

The paper proposes Un-Mixing Test-Time Normalization Statistics (UnMix-TNS), a novel BatchNorm adaptation strategy under the assumption that the test distribution is a shifting mixture. Empirical experiments on multiple CV tasks indicate that UnMix-TNS can adapt to the temporal correlation of unlabeled test streams.

**Strengths:**

Novel idea!

Intuitive toy example (i.e. Figure 1) illustrating the motivation of the proposed method.

Good empirical performance, improving over baseline methods most of the time. Mixes well (no pun intended) with other BN adaptation methods like Tent and LAME, making them even better. Synergy!

Negligible additional inference time cost (the authors should highlight it instead of burying it under a subsection titled "Ablation Studies").

**Weaknesses:**

UnMix-TNS seems tightly coupled to the convolutional layers, as it considers the channel dimension $C$ and all experiments are computer vision tasks. It is unclear whether it will work for other deep learning tasks, e.g. NLP.

The paper doesn't discuss how to choose $K$ and $\lambda$, and the additional experiments in Appendix C only study the sensitivity to $K$.

Algorithm 1 in Appendix B.2 is both helpful and confusing: helpful because it describes the algorithm concisely; but confusing because the shapes don't add up. For example, `instance_mean` should have shape `(B, 1, C)`, otherwise you cannot get `hat_mean` of size `(B, K, C)` through broadcasting. It would be better to include a runnable Python snippet in the Appendix, even if that's slightly more verbose.

**Questions:**

In Appendix B.2, it's unclear what `torch.var_mean(x, dim=[2, 3])` means. Does `x` have dimension `(batch, channel, height, weight)`?

From my understanding, UnMix-TNS will fail when test samples gradually shift from one cluster to another, in which case the active $\mu_k$ would follow the sample, causing two clusters to eventually overlap. How would you prevent that?

Not really a question, but an interesting experiment is to set $K$ to the number of corruption types, and see if $p_{b,k}^t$ can recover the corruption type for each sample correctly. Maybe each cluster will correspond to the label instead of corruption. Who knows?

---

> ### Author Response · Authors · 2023-11-20
> **Thanks for your comment! (1/2)**
>
> Thank you for your constructive comments and suggestions; and they are exceedingly helpful for us in improving our paper. We have carefully incorporated them in the revised paper. In the following, your comments are first stated and then followed by our point-by-point responses.
>
> > Applicability of UnMix-TNS beyond vision tasks, e.g., NLP.
>
> Thank you for your comment regarding the integration of UnMix-TNS with convolutional layers in computer vision tasks. Our work is specifically tailored to the batch normalization layer in image classification models. While the extension of UnMix-TNS to NLP tasks is an interesting concept, it falls outside the scope of this paper. This decision is guided by the prevalent use of layer normalization in NLP, which differs from our focus on batch normalization. In summary, exploring UnMix-TNS in NLP context would require a different approach, considering the distinct normalization techniques used in NLP.
>
> > Discussion on how to choose $K$ and $\lambda$.
>
> Thank you for pointing out the need for a clearer explanation of the selection of $K$ and $\lambda$ in our paper. Regarding $K$, its choice is crucial, as emphasized in our future work. A single UnMix-TNS component can efficiently represent several similar classes (e.g., car, truck, bus), allowing for an accurate approximation of the test distribution with a lower $K$ value. This approach is particularly beneficial for datasets with distinct class similarities. However, in datasets with a large number of classes like CIFAR100-C, DomainNet-126, or ImageNet-C, aligning $K$ with the number of classes may lead to inefficiencies and an increased memory footprint. For instance, setting $K$ to 100 in CIFAR100-C tests led to reduced performance. The optimal $K$ varies with the testing scenario; larger $K$ values are necessary for mixed domains with diverse styles, while smaller $K$ values are sufficient in single or continual domains due to reduced variability. In summary, choosing $K$ in UnMix-TNS is about balancing class/domain complexity with model efficiency, and we plan to refine these guidelines further in our future research.
>
> Concerning $\lambda$: Equations (12) and (13) are derived based on the principles of momentum batch normalization (MBN), and the hyper-parameter $\lambda$ is set according to the MBN guidelines proposed by Yong et al. (2020). MBN aims to standardize the noise level introduced by Batch Normalization (BN) across different batch sizes, ensuring comparable noise levels between small and large batches. The tuning of $\lambda$ is integral to achieving this standardization based on the specific batch size employed. For a more in-depth explanation, we have added new information in Appendix A.4 of our paper.
>
> > Algorithm 1 in Appendix B.2: Concise but confusing with shape mismatch.
>
> Thank you for your valuable feedback regarding Algorithm 1 in Appendix B.2. We understand your concerns about the clarity of the shapes used in the algorithm and apologize for any confusion caused. To address this, we have revised Algorithm 1 to ensure the correct alignment of tensor shapes.
>
> In the updated algorithm, we have introduced the **unsqueeze** operation to adjust the dimensions of various tensors appropriately. The **instance\_mean** is now unsqueezed along dimension 1, altering its shape to (B, 1, C). This adjustment allows for the correct broadcasting when calculating the **hat\_mean** of size (B, K, C). Additionally, we have applied the unsqueeze operation to self.components\_means along dimension 0, resulting in a shape of (1, K, C). Furthermore, we have unsqueezed $p$ along the last dimension, changing its shape to (B, K, 1). By incorporating all the above steps, we can now correctly combine these tensors, ensuring the resultant **hat\_mean** has the intended size of (B, K, C). We hope that these clarifications and the revised algorithm will make the implementation more straightforward and eliminate any confusion.
>
> > Unclear dimension explanation in Appendix B.2 for torch.var\_mean(x, dim=[2, 3]).
>
> Thank you for your query. We would like to clarify that in this context, the variable $x$ indeed represents a batch of intermediate features within a Convolutional Neural Network (CNN). As is typical with CNNs, the tensor $x$ possesses four dimensions: batch size, number of channels, height, and width. Therefore, when we call torch.var\_mean(x, dim=[2, 3]), the function calculates the variance and mean across the height and width dimensions of the tensor $x$. This operation effectively computes the instance-wise statistics for each feature in the batch, considering the spatial dimensions (height and width) of the CNN's intermediate feature maps.

---

> > ### Author Response · Authors · 2023-11-20
> > **Thanks for your comment! (2/2)**
> >
> > > Handling gradual shift between clusters.
> >
> > Thank you for your insightful comment regarding the potential issue of UnMix-TNS when test samples gradually shift between clusters, leading to an overlap of clusters.
> >
> > We would like to highlight that such a scenario can happen if the semantics of an input image change gradually over time, causing a gradual drift in the feature representation. To address this, UnMix-TNS is designed to update the statistics of the active component slowly, using an exponential moving average. This method of updating helps in mitigating the risk of rapid shifts in the active  $\mu_k$. Moreover, by choosing a lower value for the momentum hyperparameter $\lambda$,  we can further slow down the movement of the active $\mu_k$, making it less susceptible to quickly following the distribution drift. This careful calibration ensures that the active $\mu_k$	responds to changes in the data distribution at a controlled pace, reducing the likelihood of overlapping clusters due to rapid shifts in feature representation.
> >
> > > Experiment idea: Setting clusters to corruption types in UnMix-TNS.
> >
> > Thank you for your intriguing suggestion regarding the experiment on setting $K$ to the number of corruption types. Furthermore, conducting an experiment to test this hypothesis could be very enlightening, potentially revealing how different layers of a network interpret and represent various aspects of the data, from style and domain at the initial layers to semantic content towards the end.
> >
> > In line with your suggestion, we have conducted additional experiments on CIFAR10-C under mixed-domain scenarios. We performed two sets of experiments: firstly, setting $K=15$, which corresponds to the number of corruption types, and secondly, setting $K=150$, representing the number of corruptions multiplied by the number of classes. Our findings from these experiments were quite revealing. In the first scenario, we observed an average error rate of 63.5\% across the 15 corruption types. However, in the second scenario, where $K$ was significantly higher, the error rate decreased to 40.9\%. These results underscore the importance of choosing an appropriate value for $K$ depending on the test scenario. In mixed domain scenarios, with a wide range of styles in the test distribution, a larger $K$ is essential for effectively capturing this diversity. This is in contrast to single domain scenarios, where the domain is consistent, or continual domain scenarios, with sequential domain shifts.
> >
> > In summary, our experiments reinforce the notion that selecting $K$ in UnMix-TNS is a balancing act between the complexity of the classes and domains, and the practical considerations of model efficiency. Particularly in datasets with a large number of classes, setting $K$ equal to the number of classes can lead to inefficiencies, as demonstrated by our experimental results. We believe these findings will add a valuable perspective to our understanding of how $K$ influences model performance in various scenarios.

---

> ### Comment · Reviewer_auPw · 2023-11-23
>
> Thanks to the authors for the additional clarification and experiments. Overall, I think addressing temporally correlated test data is a realistic and interesting research direction, and this paper did a good job tackling it. I have increased my rating to 8. I look forward to your future work, hopefully with real-world applications.
>
> Before going into details, I will reiterate my recommendation that you should put the phrase "with negligible inference latency" somewhere in the abstract, or maybe even do an experiment comparing the inference latency with other methods like Tent. UnMix-TNS looks very complex and use cases like autonomous driving care about inference latency a lot, so it's good to know it admits an efficient implementation.
>
> > To address this, UnMix-TNS is designed to update the statistics of the active component slowly, using an exponential moving average. This method of updating helps in mitigating the risk of rapid shifts in the active $\mu_k$.
>
> I was worried about a gradual, instead of rapid, shift in feature representation. As you pointed out, the cluster mean $\mu_k$ is pretty robust to rapid shifts thanks to the use of a moving average, but if the shift is gradual I'm not sure how a moving average helps. On the other hand, I should probably not worry too much about that, because we won't see a lot of samples sitting on the decision boundary in practice: those correspond to ambiguous samples that are information-theoretically hard to classify.
>
> > In line with your suggestion, we have conducted additional experiments on CIFAR10-C under mixed-domain scenarios.
>
> Sorry for the confusion, but what I meant is "To what extent does each instance's cluster membership correlate with its class label?". However, your result is still helpful: I can infer that the answer is probably "not much", because each class can have multiple subclasses/domains, e.g. a dog in the snow may look more like a deer in the snow than a dog on the grass.

---

> > ### Author Response · Authors · 2023-11-23
> > **Thanks for your additional comment!**
> >
> > Dear Reviewer auPw,
> >
> > We are immensely grateful for your revised rating and positive feedback on our paper. Your recognition of our work's contribution to addressing temporally correlated test data is greatly appreciated, and we are excited about the potential real-world applications of our research.
> >
> > Regarding your recommendation to highlight the "negligible inference latency" of UnMix-TNS, we find this suggestion highly relevant, especially for applications where inference speed is crucial. To this end, we included this phrase in the contribution list (third row) in the revised manuscript for clarity, as updating the abstract is not feasible on the OpenReview at this stage. Additionally, our experimental findings on inference latency reveal the following:
> >
> > * Using the Source model, the latency is 0.48 milliseconds per sample.
> > * Implementing UnMix-TNS results in a slightly higher latency of 0.63 milliseconds per sample, still maintaining a rapid processing time.
> > * The TENT method shows an increased latency of 0.97 milliseconds per sample.
> >
> > These results demonstrate that while UnMix-TNS introduces a slight increase in latency, it remains significantly efficient.
> >
> > Your concern about the potential gradual shift in feature representation is valid. The moving average helps in slowing down the UnMix-TNS components, and by choosing a lower value for the momentum hyperparameter, we can further slow down the movement of the active $\mu_k$ (slower than the shifts in features), making it less susceptible to potential gradual shift. Nonetheless, as you noted, instances that sit on the decision boundary, representing ambiguous samples, are relatively rare in practice and in our used datasets. This reduces the likelihood of encountering significant issues due to gradual shifts. Nonetheless, this is an important aspect to monitor in future applications and research.
> >
> > In response to your query about the correlation between an instance's cluster membership and its class label, we agree that our results suggest a weak correlation. As you pointed out, within a single class, there can be multiple subclasses or domains, leading to a diverse range of features within the same class. This diversity reinforces the complexity of the problem and the necessity of approaches like UnMix-TNS that can handle such variations effectively.
> >
> > Thank you for your insightful feedback and constructive suggestions, which have greatly refined our research. We are committed to further exploring and applying our work and addressing the challenges you've highlighted. We look forward to advancing this field of research.

---

### Official Review · Reviewer_Yi31 · 2023-11-10

**Soundness:** 2 fair
**Presentation:** 3 good
**Contribution:** 2 fair
**Rating:** 6
**Confidence:** 3

**Summary:**

This paper studies how to adapt the batch norm layers for test-time adaptation. Standard TBN method assumes that test samples are i.i.d. sampled from a single target distribution, while this assumption can be violated in real scenarios: samples can be drawn from multiple distributions and can be temporal correlated. The author propose a nuanced method called UnMix-TNS, which split the running statistics of each BN layer into K different components. For each testing sample, only the closest component will be refined, which makes the BN layer more stable and robust to changes in batch-wise label distribution. The author verify the proposed method over benchmarking datasets, several settings, a wide range of models, and compare to a variety of baselines.

**Strengths:**

1. Clear motivation: The author clearly summarizes key drawbacks of BN-based TTA methods. Figure 1 is intuitive and informative.
2. Algorithm design: The algorithm is generally intuitive to me. Clearly if each component corresponds to a label, the algorithm is likely to be robust to the bias in label distribution. Also the author pay special attention on the initialization of UnMix-TNS components to preserve the statistical properties.
3. Experiments: The proposed method is tested on a variety of datasets, models, evaluation protocols, and is compared to a wide range of related baselines.

**Weaknesses:**

Major (These weaknesses or concerns significantly affect my understanding and decision regarding this paper)

1. Multiple target domains: One of the claimed contribution is that UnMix-TNS has robustness when tested on continual domain and mixed domain. However, I am very confused which part of UniMix-TNS is designed for and beneficial to these multi-domain settings, especially the mixed domain setting. Considering that there is a new batch containing images from different domains, although different images may correspond to different UnMix-TNS components, they are finally normalized with the same mean and variance according to Eq (11). It seems like UnMix-TNS still treat multiple target domains as one single target domain.
2. Unclear experimental setting and unsatisfactory performance: The author claimed that they follow the protocol outlined in (Lim et al., 2023). However, the performance for most of the baselines are significantly worse than the results in (Lim et al., 2023). Also, the proposed UnMix-TNS fails to outperform “Source” in mixed domains. I believe the comparison to baselines only makes sense if the test-time normalization is beneficial. (I presume that “Source” means no adaptation. Please correct me if I am wrong)
3. If the temporal correlation mainly
4. Choice of K. The appendix discusses the influence of K. However, it is still unclear how to choose K in practice. How is it related to the number of classes and number of domains? (Although the number of domains might not be exposed.)

Minor (These minor weaknesses are not crucial but I believe fixing them will improve the quality of the paper)
1. Figure 2 is pixelated. Please consider improving the dpi.
2. The temporal correlation might be better explained in Section 2.1. Does it refer to correlation of feature, label, or domain?
3. Page 5 after Equation (1), two exp have different font style. Also I recommend changing $\sim$ to $\approx$ since $\sim$ usually means “following the distribution of”.

**Questions:**

Besides the major weaknesses, there are several minor questions:
1. In Equation (5), it seems like all $\mu_{k, c}^0$ for different $k$ distribute on the line of $\mu_c + t \sigma_c$. Is that intentional? Or it makes more sense if they do not have such low-rank structure?
2. Equation (7). Are there any insight on using cosine similarity instead of L2 distance?

**Details Of Ethics Concerns:**

I do not have any ethics concerns.

---

> ### Author Response · Authors · 2023-11-20
> **Thanks for your comment! (1/2)**
>
> Thank you for your constructive comments and suggestions. We have carefully addressed them. In the following, your comments are first stated and then followed by our point-by-point responses.
>
> > How does UnMix-TNS effectively handle mixed domain settings when its normalization suggests it treats multiple domains as a single one?
>
> Thank you for your insightful comment. We agree that UnMix-TNS does not incorporate a specific module tailored for mixed domain settings. However, we want to shed light on the fact that UnMix-TNS shows robustness against temporal correlations (different values of Dirichlet coefficient $\delta$) of the test images with respect to their inaccessible labels compared to other test time normalization schemes.
>
> The core functionality of UnMix-TNS lies in its ability to retain current and past statistics of online test features through its $K$ components, functioning as a time-decaying memory. In cases where test features are label-correlated, UnMix-TNS effectively simulates an i.i.d. scenario. It accomplishes this by estimating the current mean and standard deviations from its $K$ components, each representing different groups of test features encountered previously.
>
> Additionally, the gradual update of the $K$ components over time enables UnMix-TNS to adeptly handle continual domain settings, where domain shifts occur sequentially rather than randomly. This capability was demonstrated in our experiments, showing UnMix-TNS's effectiveness not only in single domain settings but also in continual domain scenarios. Therefore, while UnMix-TNS may not specifically target mixed domain settings, its underlying design principles and operational mechanics provide flexibility and adaptability across varying domain configurations, including mixed domain. Furthermore, as supported by our experiments, our method's adaptability and performance in mixed-domain scenarios can be further enhanced by adjusting the momentum hyperparameter $\lambda$ and appropriately increasing the value of $K$ to encompass the style diversity in the mixed domain.
>
>  Finally, our claim regarding the mixed domain scenario has been revised in the updated manuscript.
>
> > Unclear experimental setting, with UnMix-TNS not surpassing "Source" in mixed domains and baseline performances falling behind those in Lim et al. (2023)
>
> Thank you for your comment and for highlighting these concerns. Firstly, we apologize for any confusion caused by the experimental protocol reference. We followed the protocol set out in Marsden et al., 2023, not Lim et al., 2023. This error has been corrected in our revised manuscript.
>
> Regarding the performance of baselines, it's important to note that Lim et al., 2023, primarily reported baseline performances in an i.i.d. setting. They only report the performance of their method for class imbalance for the single domain setting. Our approach with UnMix-TNS, while occasionally not surpassing the source model, does show consistent improvements over other test-time normalization schemes, particularly where these schemes may not have yielded significant enhancements. Additionally, it's important to recognize that our results are consistent with Lim et al., 2023's observation. They noted underperformance compared to the source model in class imbalance settings and indicated that relying solely on normalization techniques may not always lead to significant improvements in non-i.i.d settings.
>
> However, UnMix-TNS stands out by offering notable improvements to test-time adaptation methods. In some cases, it even helps these methods outperform the source model in different scenarios, including mixed domain. For instance, when integrating UnMix-TNS with ROID, we observed a significant performance leap over the source model, a feat not achieved with TBN as the normalization technique. It is noteworthy that UnMix-TNS outperforms the source model in the mixed domains scenario on DomainNet-126 and CIFAR10-C datasets. While the results for mixed domains have been reported with $K=128$ in the manuscript, increasing $K$ to $K=1024$ shows even more impressive results on CIFAR100-C. Specifically, the error rate on CIFAR100-C decreases to 45.8\%, outperforming the source model by 0.7\%.
>
> Nevertheless, we can enhance the performance of UnMix-TNS in mixed domain adaptation scenarios by tuning the $\lambda$ hyperparameter. As detailed in Appendix A.4, this adjustment, guided by momentum batch normalization (MBN) principles, ensures consistent noise levels across different batch sizes. $\lambda$ is calculated as $\lambda=1-(1-\lambda_0)^\frac{B}{B_0}$ where $\lambda_0$ and $B_0$ represent the ideal values, and $B$ is the actual batch size. In mixed domain scenarios, with more varied styles and higher noise, we suggest a modified formula $\lambda=1-(1-\lambda_0)^\frac{B}{B_0*15}$ considering $15$ as the number of corruptions. This adjustment lowered the error rate on CIFAR100-C to 45.9\%, slightly outperforming the source model.

---

> ### Author Response · Authors · 2023-11-20
> **Thanks for your comment! (2/2)**
>
> > If the temporal correlation mainly...
>
>  It seems the point about temporal correlation was not completed. Could you please elaborate on this, so we can address it more effectively?
>
> > Choice of $K$
>
> We appreciate your inquiry regarding the selection of $K$ in our UnMix-TNS framework. The determination of $K$ is a critical element, as we have previously mentioned in our discussion of future work. To begin, we highlight that a single UnMix-TNS component can sometimes effectively represent multiple similar classes (e.g., car, truck, bus). This means that even with a relatively low $K$ value, our model can approximate the mean and variance of the test distribution accurately. This approach is particularly beneficial when class similarities are pronounced. However, in scenarios with a large number of classes, such as CIFAR100-C, DomainNet-126, or ImageNet-C, setting $K$ equal to the number of classes can lead to inefficiencies. Specifically, it can result in an excessive memory footprint, which is impractical for many applications. For instance, in our experiments with CIFAR100-C, we observed that setting $K$ to 100 actually decreased performance.
>
> Furthermore, the optimal value of K varies depending on the test scenarios. In mixed domain scenarios, where there is a diverse range of styles in the test distribution, a larger $K$ is necessary to capture this diversity effectively. This contrasts with single domain scenarios, where the domain remains consistent, or continual domain scenarios, where the domain shifts sequentially. In these cases, a smaller $K$ may be sufficient due to the reduced variety in the test distribution.
>
> In summary, the choice of $K$ in UnMix-TNS is a balance between the complexity of the classes and domains and the practical considerations of model efficiency. While a higher $K$ may offer more nuanced representation in diverse settings, it also comes with trade-offs in terms of computational and memory requirements. Our ongoing research aims to refine these guidelines further and provide more concrete recommendations in various application contexts.
>
> > Minor Weakness
>
> Thank you for your feedback regarding minor weakness. We acknowledge the pixelation issue of Figure 2 and have improved the dpi to enhance its clarity in the revised manuscript. Also, in the revised manuscript (Section 2.1, second paragraph, Test-time adaptation under label temporal correlation), we have clarified that it specifically refers to the temporal correlation of labels.  We thank you for noting the font style inconsistency after Equation (1) on Page 5; we corrected this in the revised manuscript. Also, we appreciate your suggestion to replace $\sim$ with $\approx$ for clarity in denoting approximation rather than distribution. This change is implemented to enhance the precision of the expressions in the paper.
>
> > Equation (5)
>
> In response to the reviewer's comment on Equation (5) regarding the distribution of $\mu_{k,c}^0$, we would like to clarify that $\mu_{k,c}^0$ is a scalar quantity representing the mean of the $k^{th}$ UnMix-TNS component for channel $c$, and $t$ is a scalar sampled from a Normal Distribution $\mathcal{N}(0,1)$. Given this, the vector $\mu_k$ does not lie on a line and has no low-rank structure.
>
> > Using cosine similarity instead of L2 distance?
>
> We thank the reviewer's comment. The L2 distance is susceptible to the curse of dimensionality, and as the number of channels increases, the contrast between L2 distances ($b^{th}$ instance's statistics and the $K$ statistic components) tends to diminish. Consequently, it becomes necessary to decrease the temperature $\tau$ when computing the assignment probability $p_{b,k}$ in Equation (10) to avoid updating all $K$ components with the same weight. Such a distance function, therefore, requires careful setting of the temperature for every layer. On the other hand, cosine similarity does not suffer from this problem and only requires fixing one unique and shared temperature for all layers. Thus, we have opted to retain this score.

---

> > ### Comment · Reviewer_Yi31 · 2023-11-21
> > **Thanks!**
> >
> > Thanks for your clarification!
> >
> > - Mixed domain: I appreciate your clarification and revision on the manuscript. I do believe this is an interesting topic for further research, which is beyond the scope of this paper.
> > - Experimental setting: Thanks for your clarification!
> > - Temporal correlation: I apologize for incompleted comments. I was wondering how temporal correlation is defined. Since it seems like it is mainly about temporal correlation on labels, I believe further comparison with Test-Time Prior Adaptation [1] methods will significantly improving the soundness of this paper.
> > - Choice of $K$: Thanks for your discussion. I still believe that an empirical way to choose K will be beneficial. Just as you said, an impropriate choice of $K$ can decrease the performance. If we do not have any method to avoid that, the proposed algorithm will not be very practical.
> >
> > In a butshell, thanks for your rebuttal! I apologize again for incomplete comments. In general, I think this is a paper with interesting idea and solid results. But I still believe the points above may contribute to the quality of this paper. I keep my score of 6 at this stage. Good luck!

---

> ### Author Response · Authors · 2023-11-22
> **Thanks!**
>
> Dear Reviewer Yi31,
>
> Thank you for your thoughtful comments. We appreciate your recognition of our revisions and value your suggestions for further enhancing our manuscript.
>
> **Mixed Domain**: We are pleased our clarification on the mixed domain aspect was well-received.
>
> **Experimental Setting**: We appreciate your acknowledgment of the clarification we provided on the experimental setting.
>
> **Temporal Correlation**: Regarding UnMix-TNS, our aim was to ensure robustness against temporal correlations (different values of Dirichlet coefficient $\delta$) of the test images concerning their labels compared to other test time normalization schemes. If your reference to Test-Time Prior Adaptation methods pertains to label distribution shifts between source and target domain, this indeed offers a relevant yet distinct scenario from our current focus. We note your suggestion for further comparison in this area. **However, the specific Test-Time Prior Adaptation method you referred to was not cited**. It would be beneficial if you could provide more clarity on the approach. While we see the value in such a comparison, we must consider the scope and current focus of our manuscript to determine the feasibility of including this comparison. If not feasible at this stage, this will certainly be a consideration for our future work to enhance the robustness of our research.
>
> **Choice of K**: We acknowledge your concerns about empirically selecting K in our clustering algorithm. While our ablation studies and further investigation and discussion have touched upon this, we appreciate your suggestion for further refinement and will continue to explore this in future work to enhance our algorithm's practicality and applicability.
>
> Thank you once again for your valuable input and for your acknowledgment of the interesting idea and solid results presented in our paper.

---

> > ### Comment · Reviewer_Yi31 · 2023-11-22
> > **Sorry for missing citation**
> >
> > Here is the survey paper I mentioned. You may refer to section 6. This direction has been studied for decades, and I believe a comparison to these methods can improve the soundness of your paper.
> >
> > [1] Jian Liang, Ran He, Tieniu Tan. A Comprehensive Survey on Test-Time Adaptation under Distribution Shifts.

---

> ### Author Response · Authors · 2023-11-22
> **Addressing New Comment on Prior Adaptation**
>
> Thank you for directing our attention to the survey paper. We acknowledge the importance of comparing our work to the methods outlined in this comprehensive survey. However, implementing a direct comparison with these methods would require installing dependencies and running code from numerous online repositories, a task that is not feasible within the current discussion period's timeframe. Despite this, we have taken steps to demonstrate the versatility and robustness of our UnMix-TNS in the Test-Time Prior Adaptation scenario.
>
> One of our top-performing test-time adaptation baselines, ROID, incorporates an additional prior correction module during test-time (as detailed in Section 4.3 of their paper). This module is designed to adjust for prior shift based on the class distribution within a batch. We conducted an ablation study where we disabled their prior correction module (ROID-(Prior Correction)) and replaced its normalization layers with our proposed UnMix-TNS layer (ROID-(Prior Correction)+UnMix-TNS). The results, as shown in the table below (second row), indicate a significant performance gain over the ROID method without prior correction, even surpassing the performance of ROID with the prior correction module (ROID+(Prior Correction)) (third row). Furthermore, adding our normalization layer to the standard ROID (last row in the table) even further enhances the performance. These findings underscore the robustness of UnMix-TNS without the need for an additional prior correction module.
>
> This compelling result motivates us to consider integrating UnMix-TNS into other related test-time adaptation methods in future work in scenarios slightly different but related to Test-Time Prior Adaptation.
>
> We hope this additional insight and the efforts we have made in this direction positively influence your final rating and demonstrate our commitment to advancing the field of Test-Time Adaptation.
>
> [1] Robert A Marsden, Mario Dobler, and Bin Yang. Universal test-time adaptation through weight ensembling, diversity weighting, and prior correction. arXiv preprint arXiv:2306.00650, 2023.
>
> | METHOD      | GAUSS | SHOT | IMPUL. | DEFOC. | GLASS | MOTION | ZOOM | SNOW | FROST | FOG | BRIGH. | CONTR. | ELAST. | PIXEL | JPEG | AVG. |
> | ----------- | ----------- | ----------- | ----------- | ----------- | ----------- | ----------- | ----------- | ----------- | ----------- | ----------- | ----------- | ----------- | ----------- | ----------- | ----------- | ----------- |
> ROID-(Prior Correction) | 77.4 | 76.9 | 80.0 | 73.6 | 80.2 | 73.8 | 73.8 | 75.3 | 73.9 | 73.0 | 71.6 | 73.2 | 77.3 | 76.4 | 78.6 | 75.7 |
> **+UnMix-TNS** | 39.5 | 37.9 | 47.6 | 21.3 | 44.1 | 20.8 | 19.9 | 24.8 | 24.7 | 21.9 | 13.1 | 18.9 | 33.0 | 32.5 | 37.8 | 29.2 |
> ROID+(Prior Correction) | 75.6 | 74.9 | 78.7 | 70.7 | 79.2 | 71.1 | 71.1 | 72.7 | 71.2 | 70.0 | 68.5 | 70.2 | 75.7 | 74.2 | 76.8 | 73.4 |
> **+UnMix-TNS** | 24.4 | 22.5 | 32.3 | 8.4 | 27.9 | 7.7 | 7.6 | 10.3 | 11.2 | 9.4 | 3.9 | 7.3 | 17.1 | 17.7 | 21.7 | 15.3 |

---

### Meta-Review · Area_Chair_x3YL · 2023-12-09

**Metareview:**

Test-time adaptation (TTA) updates a model during inference, and updating normalization statistics, such as those of batch norm, is a common component of such updates. While the purpose of such methods is to cope with test data subject to shift, many methods explicitly or implicitly assume that test data still arrives in batches that are i.i.d. w.r.t. the label distribution, and so when this assumption does not hold the statistics updates can fail to yield improvement or can even backfire and reduce accuracy. The proposed Un-Mixing Test-Time Normalization (UnMix-TNS) extends these statistics updates to make them more effective on dependent orders of the data, mixed domain shifts, and continual domain shifts. The extension is essentially a mixture model, where k components (with k a chosen hyperparameter) are updated across batches and assigned to each data point within a batch. UnMix-TNS is compatible with a variety of adaptation methods by its focus on the statistics update that is a component of many of them. This variety is covered by the experiments across four normalization baselines and six optimization baselines on standard corruption benchmarks (ImageNet-C, CIFAR-10/100-C) and a natural shift benchmark (DomainNet) in multiple settings (single, continual, and mixed) in non-i.i.d. evaluations. Unlike existing methods for the non-i.i.d. setting, UnMix-TNS does not require a memory/replay buffer to simulate i.i.d. sampling (like NOTE and RoTTA), but it still improves in this setting. Although improvement is not universal, in most cases and by a large margin (>10 points absolute) UnMix-TNS reduces error in its non-i.i.d. evaluations.

All reviewers are positive but vary in their support from accept (8) to marginally above (6, 6, 6). The area chair sides with acceptance due to the extensive experimental results across multiple kinds of shifts and non-i.i.d. settings. However, this is conditional on including the missing prior papers on mixture modeling for normalization (see "Justification for Why Not Higher Score").

**Justification For Why Not Higher Score:**

Though novel to test-time adaptation, UnMix-TNS has close connections to existing work, and the related work on normalization is uncredited. The missing related work on mixture modeling for batch statistics and normalization deserves citation for the use of K components [A, B]. Furthermore, there is a close relationship between the proposed UnMix-TNS and IABN from NOTE (Gong et al. NeurIPS'22), although the proposed method differs in its mixture modeling of the statistics. Lastly, the scope of this paper is vision alone although adaptation could serve other domains.

[A]. Training faster by separating modes of variation in batch-normalized models. Kalayeh & Shah. PAMI'19.
[B]. Compound Batch Normalization for Long-tailed Image Classification. Cheng et al. MM'22.

**Justification For Why Not Lower Score:**

Test-time adaptation (TTA) is a burgeoning topic, and this submission engages with a current wave of work that examines the difficulty of adapting to streams of dependent data and proposes solutions to maintain or improve generalization in such settings. Unlike successful prior work on this front (NOTE and RoTTA), this submission takes a different path and does not make use of a replay buffer, but instead maintains more statistical "state" across inputs in the form of its mixture components. The thorough coverage of gold standard benchmarks for TTA on images (ImageNet-C, CIFAR-10/100-C) alongside other shift benchmarks like natural shift and video exceed the bar for test-time adaptation papers which sometimes only cover one such type of benchmark. Rejecting this submission would fail to inform the community about this alternative approach to adaptation across correlated inputs and perhaps prevent even more extensions given the wide compatibility of the proposed method with our TTA methods for vision.

---

### Decision · Program_Chairs · 2024-01-16

Accept (poster)